# A Studentized Spherical Harmonics–Based Nonparametric Two-Sample Test for Compositional and Directional Data

Binglin Li [1]   Matthew Reed [2]   Seong-Tae Kim [1]

## Abstract

Compositional data analysis has gained increased attention due to the widespread occurrence of simplex-valued data, including microbiome data and financial portfolios. Existing compositional two-sample tests often require log-transformations and only detect mean differences, motivating the need for a more general framework without relying on log-based methods. There is a close connection between compositional data and directional statistics, and we construct a unified non-parametric two-sample test framework. Our work is based on a studentized energy statistic constructed from spherical harmonics theory over a fixed dimensional underlying space, incorporating U-statistics theory and recent developments of studentization for both compositional and directional data. We establish asymptotic normality for our spherical harmonics based test statistics, thus avoiding the need for permutation tests or bootstrap procedures. Our proposed framework sheds new light on the connections between Non-Euclidean data analysis and classical asymptotic high-dimensional data techniques.

## 1. Introduction

Two sample tests have a long history in statistics, however, new challenges always arise when data are non-Euclidean. Compositional data, originally modeled on a simplex, has recently been revealed to be projective in nature, and bears a deeper connection with directional statistics (Li et al., 2023). Developing nonparametric tests, e.g., two-sample tests and goodness-of-fit test, for compositional and directional data, which share a common spherical geometric foundation, both

fall under "Non-Euclidean Nonparametric Tests."

In compositional two-sample test, starting from Aitchison's work back in the 80s, log-transformations have been a dominant approach in compositional data analysis (CoDA) (Greenacre et al., 2023), and as a result, the majority of the previous works on CoDA have adopted Aitchison's log-transformations, e.g., (Cao et al., 2018; Zhang & Dao, 2020). However, an obvious caveat of the log-transformation approach is the existence of zero-valued components in compositional data, especially in high-dimensional data such as human microbiome data (Weiss et al., 2017). To avoid this issue, many studies in CoDA still assume strictly positive components, rather than addressing zero values (Zhang et al., 2025; Li et al., 2025). Therefore, new approaches for CoDA without log-transformations in the presence of zero values are highly desirable.

The key drawback of log-based methods is that they reinforce linear structures on simplices by deleting their boundaries, which causes many issues both in theory and in practice; but modern machine learning has developed rich kernel techniques to linearize nonlinear spaces. In light of close connections between compositional domains and spherical domains (see Scealy & Welsh (2011) and Li et al. (2023)), compositional data analysis should be Non-Euclidean in nature; hence, leveraging kernel techniques should be a natural and modern choice.

While Li et al. (2023) used the reproducing kernel Hilbert space (RKHS) and $\Gamma$-invariant spherical harmonics for compositional data without any need of log-transformations or any fear of boundary issues, our framework is significantly different from theirs. In two sample test problems, the presence of boundaries is not an issue, because a compactly supported function on the first orthant sphere $\mathbb{S}_{\geq 0}^d$ is still a function class in $L^2(\mathbb{S}^d)$. Therefore, it is enough to directly utilize spherical harmonics theory on the entire sphere for both compositional and directional two-sample problems, and Section 3.1 further explains that the $\Gamma$-invariant approach in Li et al. (2023) is unnecessary[1], hence a uni-

[1]Department of Mathematics & Statistics, North Carolina A&T State University, Greensboro, NC 27411, United States [2]Deep AI, Laguna Beach, CA 92651, United States. Correspondence to: Seong-Tae Kim <skim@ncat.edu>.

*Proceedings of the 43ʳᵈ International Conference on Machine Learning*, Seoul, South Korea. PMLR 306, 2026. Copyright 2026 by the author(s).

---

[1]Even worse, the compositional kernels in Li et al. (2023) cannot be used to studentize our test statistic, due to their slow growth rate with respect to the degrees.

fied framework for both compositional and directional two-sample problems can be developed on the unit sphere $\mathbb{S}^d$ alone by utilizing the function space $L^2(\mathbb{S}^d)$.

The whole Hilbert space $L^2(\mathbb{S}^d)$ does not have reproducing kernels, but spherical harmonics theory tells us that each direct summand $\mathcal{H}_i$ in the Laplacian eigen-decomposition $L^2(\mathbb{S}^d) \cong \bigoplus_{i=0}^{\infty} \mathcal{H}_i$ is a finite-dimensional RKHS, thus we can apply kernel techniques on those reproducing kernel Hilbert spaces $\{\mathcal{H}_i\}_{i \geq 0}$ to study data problems on $\mathbb{S}^d$.

Despite many successful applications of kernel methods in two-sample tests, one quickly runs into difficulties when directly applying Euclidean constructions to compositional/spherical settings, because all existing applications assume that reproducing kernels are characteristic (Fukumizu et al., 2009; Sriperumbudur et al., 2009); namely, the map in (1) is an embedding from probability distributions into the Hilbert space:

$$\mathbb{P} \mapsto \mu_{\mathbb{P}} := \int_{\mathcal{X}} k(x, \cdot) \mathrm{d}\mathbb{P}. \tag{1}$$

By contrast, *none* of the reproducing kernels from $\{\mathcal{H}_i\}_{i \geq 0}$ is characteristic, therefore, any constructions based on kernel mean embeddings (Muandet et al., 2017) *cannot* be directly applied.

However, spherical harmonics theory provides infinitely many reproducing Hilbert spaces $\{\mathcal{H}_i\}_{i \geq 0}$ for a fixed unit sphere $\mathbb{S}^d$, so in order to resolve this "non-characteristic issue," we take into account those reproducing Hilbert spaces for $\mathbb{S}^d$ *all together* for the construction of our test statistics. In order to control the wild behavior of the statistics constructed from *all* those Hilbert spaces, we "studentize[2]" this statistic to obtain asymptotic Gaussian behavior under null hypotheses. Our contributions can be summarized as below:

- ✍ We propose a fully nonparametric two-sample testing framework for compositional data *without* log-ratio transformations or any special treatment on zero-valued components, by reformulating CoDA as a directional statistics problem on spheres. As a result, our construction gives a unified framework for nonparametric two-sample tests for both compositional and directional data.
- ✍ Based on the theory of spherical harmonics, we construct a kernel energy distance (KED) on $H_p := \bigoplus_{i=0}^{2p} \mathcal{H}_i$ for each finite level "$p$", and take all finite levels together into consideration to distinguish large classes of probability measures. Furthermore, we studentize these spherical harmonic energy statistics to obtain an asymptotically well-behaved statistic under the null hypothesis, proving a CLT, which completely

---

[2]See Section 5 for more details.

avoids permutation tests or wild-bootstrap tests as in Gretton et al. (2012).
- ✍ Simulation studies on both compositional and directional data show that our studentized test outperforms competing methods. One scenario where our method shines is when the means agree and the distributional differences occur in higher-order moments.

## 2. Problem Setup

**Assumption 2.1.** Let $(\mathcal{X}, \mathcal{B})$ be a measurable space. Let $P$ and $Q$ be probability measures on $\mathcal{X}$. We observe two samples, each consisting of independent and identically distributed (i.i.d.) observations:

$$X_1, \ldots, X_m \overset{\text{i.i.d.}}{\sim} P, \qquad Y_1, \ldots, Y_n \overset{\text{i.i.d.}}{\sim} Q.$$

In this article, the underlying space $\mathcal{X}$ will be a unit sphere $\mathbb{S}^d := \{(x_1, \ldots, x_{d+1}) | \sum_{i=1}^{d+1} x_i^2 = 1\}$ inside $\mathbb{R}^{d+1}$, and we consider the following *non-parametric* two-sample testing problem on the unit sphere $\mathbb{S}^d$

$$H_0: \ P = Q \qquad \text{v.s.} \qquad H_1: \ P \neq Q, \tag{2}$$

where $P$ and $Q$ are probability measures on $\mathbb{S}^d$.

Based on the above problem setup, compositional two-sample problems do not seem to fall within our scope. However, it is well known that compositional data analysis has close connections to directional statistics. In particular, compositional two-sample problems can be completely reduced to directional two-sample problems on $\mathbb{S}^d$, which we will elaborate on in Section 3.

## 3. Unification of Compositional & Directional Two Sample Problems

There has been a long history of connecting compositional data with directional statistics on spheres; to date, there have been a few ways to relate composition domains to $\mathbb{S}^d$, e.g., square-root transforms from $\Delta^d$ to $\mathbb{S}^d_{\geq 0}$, and a more recent approach in Li et al. (2023), which re-interpreted compositional data problems as $\Gamma$-invariant directional statistics.

In this article, we will argue that the viewpoint of Li et al. (2023) is completely unnecessary for two sample problems (see Section 3.1), and instead will directly relate the compositional simplex $\Delta^d$ to $\mathbb{S}^d_{\geq 0}$, although not necessary via square-root transformations (see Section 3.2).

### 3.1. Abandoning $\Gamma$-invariant Approach

A key theoretical insight in Li et al. (2023) was to establish an equivalence between CoDA and $\Gamma$-invariant directional statistics. To achieve this equivalence, the "spread-out" construction is the key step, which translates a compositional

dataset into a directional data set on $\mathbb{S}^d$ with $\Gamma$-symmetry, after applying a stretching map from $\Delta^d$ to $\mathbb{S}^d_{\geq 0}$. The details of the spread-out construction can be found in Section 2.1 of (Li et al., 2023), but the basic idea is illustrated in Figure 1.

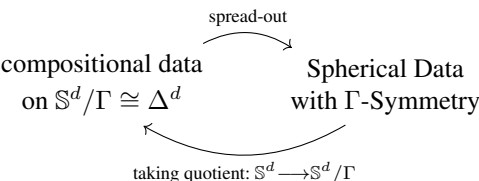

*Figure 1.* Equivalence between compositional data and $\Gamma$-symmetric spherical data.

However, a major concern is that the reflection group (in the spread-out process) $\Gamma \cong \prod_{i=1}^{d+1} \mathbb{Z}/2\mathbb{Z}$ contains $2^{d+1}$ elements. After applying the spread-out procedure, the sample size will increase from $n$ to $2^{d+1} \times n$, which poses challenges for large dimensions $d$.

We contend that such a spread-out construction is not necessary for the purpose of two-sample problems. Indeed, a compositional dataset can be viewed as data on the first-orthant sphere $\mathbb{S}^d_{\geq 0}$ whose probability measure $P$ is only supported on $\mathbb{S}^d_{\geq 0}$, then a spread-out procedure creates a distribution which "replicates" the original distribution $P$ on $\mathbb{S}^d_{\geq 0}$ in every other orthant of $\mathbb{S}^d$ up to a constant number. More specifically, assume that $P$ admits a density $\mathfrak{p}$ on $\mathbb{S}^d_{\geq 0}$, then we can extend this density to a function $\tilde{\mathfrak{p}}$ on the whole sphere $\mathbb{S}^d$ by setting it to zero outside the first orthant:

$$\tilde{\mathfrak{p}}(x) = \left\{ \begin{array}{ll} \mathfrak{p}(x), & x \in \mathbb{S}^d_{>0} \\ 0, & x \in \mathbb{S}^d \setminus \mathbb{S}^d_{\geq 0} \end{array} \right.$$

then after this extension, the spread-out data has the following density $\mathfrak{p}^\Gamma$:

$$\mathfrak{p}^\Gamma(x) = \frac{1}{|\Gamma|} \sum_{\gamma \in \Gamma} \tilde{\mathfrak{p}}(\gamma x), \quad x \in \mathbb{S}^d. \tag{3}$$

Then an easy observation leads to the following equivalence:

$$P \neq Q \Leftrightarrow \tilde{P} \neq \tilde{Q} \Leftrightarrow P^\Gamma \neq Q^\Gamma, \tag{4}$$

where $P, Q$ are probability measures supported on $\mathbb{S}^d_{\geq 0}$, $\tilde{P}, \tilde{Q}$ denote their zero extensions to $\mathbb{S}^d$, and $P^\Gamma, Q^\Gamma$ denote their $\Gamma$-symmetrizations on $\mathbb{S}^d$.

The three equivalences in (4) are stated only at a logical equivalence. Recall the assumption for this subsection that the distributions in (4) admit densities, $\mathfrak{p}, \mathfrak{q}; \tilde{\mathfrak{p}}, \tilde{\mathfrak{q}}$, and $\mathfrak{p}^\Gamma, \mathfrak{q}^\Gamma$, which can be viewed as both compactly supported functions and densities on $\mathbb{S}^d$. Since $L^1$-norms are frequently used to measure distances between densities, we can upgrade

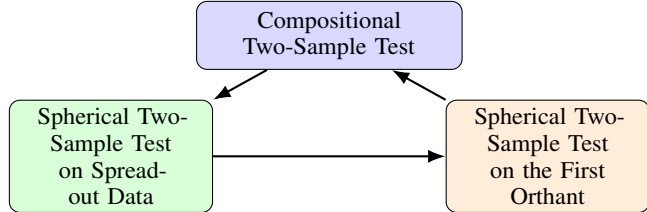

*Figure 2.* Equivalence of two-sample tests for compositional data in the three geometric contexts.

the logical equivalence in (4) to the following equalities of $L^1$-distances:

$$\|\mathfrak{p} - \mathfrak{q}\|_1 = \|\tilde{\mathfrak{p}} - \tilde{\mathfrak{q}}\|_1 = \|\mathfrak{p}^\Gamma - \mathfrak{q}^\Gamma\|_1. \tag{5}$$

Identity (5) indicates that it suffices to distinguish two spherical distributions $\tilde{P}$ and $\tilde{Q}$ *without* the need for spread-out constructions. Thus, we summarize the equivalence of three different scenarios of two-sample tests in Figure 2.

### 3.2. From $\Delta^d$ to $\mathbb{S}^d_{\geq 0}$ with Flexibility

All discussions in Section 3.1 assume a map from a compositional simplex $\Delta^d$ to the unit sphere $\mathbb{S}^d_{\geq 0}$; traditionally such a map is obtained through square-root transformations. However, a new suggestion of such a map was given by "stretching" (see Figure 1 in (Li et al., 2023) for an intuitive illustration).

Stretching is certainly better for compositional dimension reduction problems, but either map (stretching or square-root transformations) is sufficient for two sample problems, which gives readers flexibility of choices.

Therefore, compositional two-sample problems can be viewed as special cases of directional two-sample problems after choosing a transformation from $\Delta^d$ to $\mathbb{S}^d_{\geq 0}$.

## 4. Spherical Energy Distances at Finite Levels

In Section 3, we explained how compositional two-sample problems can reduce to directional two-sample problems on $\mathbb{S}^d$. Hence, it suffices to construct a unified nonparametric two-sample test framework on the unit sphere $\mathbb{S}^d$, which includes compositional tests as special cases.

### 4.1. Spherical Kernels from Spherical Harmonics theory

Directional two-sample problems fall into the realm of Non-Euclidean Data Analysis. It is generally challenging to recover high dimensional frameworks on "curved manifolds". The central philosophy in our work is to "linearize" non-Euclidean spaces by considering their function spaces, whose linear structures can be used to "revitalize" classical high-dimensional techniques in non-Euclidean contexts.

The key player for studying function spaces on spheres is spherical harmonics theory. For a fixed-dimensional unit sphere $\mathbb{S}^d$, spherical harmonics theory provides infinitely many Hilbert spaces $\{\mathcal{H}_i\}_{i \geq 0}$ with reproducing kernels, which are parametrized by degree parameters $i \in \mathbb{Z}_{\geq 0}$.

Each single $\mathcal{H}_i$ is well-understood, and their direct sums $\bigoplus_{i=0}^{n} \mathcal{H}_{2i}$ and $\bigoplus_{i=0}^{n} \mathcal{H}_{2i+1}$ can be conveniently identified with homogeneous polynomial spaces (see (24) in Appendix A.1). In this article, all statistics are constructed from the building block $H_p := \bigoplus_{i=0}^{2p} \mathcal{H}_i$[3].

## 4.2. Projected Energy Distances via Spherical Kernels

In this section, we use spherical kernels to develop KED between two distributions, $P$ and $Q$, following Assumption 2.1, and let $(X', Y')$ denote an i.i.d. copy of $(X, Y)$. The original energy distance (ED), in the sense of Székely & Rizzo (2013), is defined as:

$$\mathrm{ED}(X, Y) = 2\mathbb{E}[d(X, Y)] - \mathbb{E}[d(X, X')] - \mathbb{E}[d(Y, Y')].$$

It is also well known that ED admits a kernelized version: namely, given a characteristic kernel $k(\cdot, \cdot)$, the kernel energy distance (KED), equivalently, the squared maximum mean discrepancy of Gretton et al. (2012), is

$$\mathrm{KED}(P, Q) = \mathbb{E}[k(X, X')] + \mathbb{E}[k(Y, Y')] - 2\mathbb{E}[k(X, Y)].$$

In our situation, we can construct KED on finite-level Hilbert spaces; indeed, recall that the $p$-th finite-level Hilbert space is $H_p = \bigoplus_{i=0}^{2p} \mathcal{H}_i$, whose reproducing kernel is given by

$$K_p(x, y) := \sum_{i=0}^{2p} k_i(x, y), \quad \forall x, y \in \mathbb{S}^d. \tag{6}$$

Then the $p$-th projected KED is defined by:

$$\mathrm{KED}_p(P, Q) := \mathbb{E}[K_p(X, X')] + \mathbb{E}[K_p(Y, Y')] - 2\mathbb{E}[K_p(X, Y)]. \tag{7}$$

Classical (kernel) ED between two probability distributions, $P$ and $Q$, measures the discrepancy between them. What, then, is the $p$-th KED in (7) measuring? Let us focus on probability measures on the $\mathbb{S}^d$ that can be represented by continuous functions, so that they are contained in $\bigoplus_{i=0}^{\infty} \mathcal{H}_i$, and consider the following projection map:

$$\Pi_p : \bigoplus_{i=0}^{\infty} \mathcal{H}_i \longrightarrow \bigoplus_{i=0}^{2p} \mathcal{H}_i = H_p. \tag{8}$$

Then, for two probability distributions $P$ and $Q$, applying the projection map $\Pi_p$ in (37), we obtain two projected functions $\Pi_p(P)$ and $\Pi_p(Q)$ inside $H_p = \bigoplus_{i=0}^{2p} \mathcal{H}_i$.

---

[3]The reason we choose the direct sum $H_p = \bigoplus_{i=0}^{2p} \mathcal{H}_i$ for each $p \geq 1$ is mainly for convenience, because many properties can be uniformly discussed, independently of the parity of the integer $p$, and many arguments and proofs for our main results are much more streamlined.

**Definition 4.1.** Given a probability distribution $P$ on $\mathbb{S}^d$, we define its $p$-th projected measure (or projected distribution) as the image of $P$ under the projection map in (37).

*Remark* 4.2. One can think of the projection map in (37) as an orthogonal projection of the function $P$ into the subspace $H_p$. There is no guarantee that $\Pi_p(P)$ defines a measure on $\mathbb{S}^d$, but it still makes sense to discuss the "discrepancy" and ED of projected measures between $\Pi_p(P)$ and $\Pi_p(Q)$.

Heuristically speaking, the $p$-th projected KED in (7) measures the discrepancy for $\Pi_p(P)$ and $\Pi_p(Q)$, which is also the energy distance for two projected measures.

## 4.3. Including All Spherical Harmonics and Motivation of Studentizations

In Section 4.2, the $p$-th projected KED defined in (4.1) does not measure the full distance between $P$ and $Q$, but only the distance between the projected measures $\Pi_p(P)$ and $\Pi_p(Q)$. For a fixed pair of different distributions $P$ and $Q$, a fixed degree $p$ might not be able to distinguish them, but if we keep increasing the degree parameter $p$, eventually $\Pi_p(P) \neq \Pi_p(Q)$ for sufficiently large $p$. However, one instantly runs into this simple question: for any two distributions $P$ and $Q$ on $\mathbb{S}^d$, how can we choose the degree parameter $p$ that is large enough to have $\Pi_p(P) \neq \Pi_p(Q)$? To answer this question, the choice of such a value $p$ depends on the particular choice of $P$ and $Q$. However, for completely nonparametric tests, there is no uniformly large "$p$" for all pairs of different distributions, so we cannot rely on the finite-level projected ED in (4.1) to perform nonparametric two-sample tests. In order to distinguish arbitrary pairs of different probability distributions $P$ and $Q$ on $\mathbb{S}^d$, our strategy is to consider the Hilbert spaces of all levels:

$$\lim_{p \to \infty} H_p = \lim_{p \to \infty} \bigoplus_{i=0}^{2p} \mathcal{H}_i.\text{[4]} \tag{9}$$

As $p \to \infty$, we should obtain the following "limit":

$$\lim_{p \to \infty} \mathrm{KED}_p(P, Q) = \lim_{p \to \infty} \big( \mathbb{E}[K_p(X, X')] + \mathbb{E}[K_p(Y, Y')] - 2\mathbb{E}[K_p(X, Y)] \big) \tag{10}$$

The limit in (10), if it does exist, is indeed able to distinguish any pair of different distributions apart; however, it is not practical to work with that limit directly:

- 🌐 If we construct estimators for (10), their null distributions may be difficult to characterize, or no known distributions may be available to model the null case.
- 🌐 With unknown null distributions, a common practice is to design permutation tests or (wild) bootstrap procedures, but potential issues of inconsistency is still

---

[4]Technically speaking, the limit procedure in (9) is in fact a colimit construction in the category of topological vector spaces.

present (Hagrass et al., 2024; Key et al., 2025; Zhu & Shao, 2021).

Our proposal in this article is to take advantage of the exciting recent development of studentization techniques. Studentization provides well-understood asymptotic null distributions, completely avoiding permutation/bootstrap procedures. However, our compositional/spherical studentizations differ from existing studentization procedures:

☞ In traditional studentizations (Gao & Shao, 2023; Chakraborty & Zhang, 2021; Yan & Zhang, 2023), the *underlying spaces* for data points vary, and the dimensions of the underlying spaces go to infinity ;

☞ In the compositional/spherical context, the underlying space is the *fixed* underlying space $\mathbb{S}^d$, but the Hilbert spaces built on top of this fixed space *do vary*; these Hilbert spaces are parameterized by the degree "$p$" and this degree $p$ goes to infinity, while the dimension of the underlying space remains *constant*.

## 5. Studentizations of Spherical KED

In this section, our goal is to estimate and studentize the $p$-th kernel energy distance $\text{KED}_p$ defined in (7) using U-statistics theory. This construction allows us not only to distinguish general pairs of distributions by letting $p \to \infty$, but also to obtain a studentized statistic with an elegant asymptotic distribution under the null hypothesis.

First, let us write down the estimator for $\text{KED}_p$. This estimator, denoted by $\mathcal{E}_{m,n}^p(X,Y)$, is simply the MMD-type estimator with the kernel $K_p$ in (6) for $H_p$:

$$\mathcal{E}_{m,n}^p(X,Y) := \binom{m}{2}^{-1} \sum_{1 \le i < j \le m} K_p(X_i, X_j) \qquad (11)$$
$$+ \binom{n}{2}^{-1} \sum_{1 \le i < j \le n} K_p(Y_i, Y_j) - \frac{2}{mn} \sum_{\substack{1 \le i \le m \\ 1 \le j \le n}} K_p(X_i, Y_j).$$

In Section 4.3, we discussed why to studentize the estimator $\mathcal{E}_{m,n}^p(X,Y)$. In order to do so, we need to divide the estimator by another statistic which estimates the variance component $\text{Var}[\mathcal{E}_{m,n}^p(X,Y)]$. An important player for estimating this variance has been the "squared distance covariance" $\text{dCov}^2(X,Y)$ and its kernelization.

As expected, the connection between the kernel squared distance (KSD) covariance and $\text{Var}[\mathcal{E}_{m,n}^p(X,Y)]$ is a major theme in our studentization. In this section, we give a short review of the development of kernel squared distance covariance and state its finite $p$-th level version in our context, before carefully deriving its relation with $\text{Var}[\mathcal{E}_{m,n}^p(X,Y)]$, as preparation for our final studentization.

### 5.1. KSD Covariances at Finite Level

To date, there has been a small industry of studying the estimation of "variances of energy distances." A major candidate is the famous *squared distance covariance*. Squared distance covariances were originally constructed for Euclidean vector spaces (Székely et al., 2007), and Lyons later extended the original Euclidean construction to general metric spaces in Lyons (2013). The kernelized version of distance covariance is also well known as follows with $\{(X_i, Y_i)\}_{i=1}^3$ i.i.d. copies of $(X,Y)$:

$$\mathcal{V}_k^2(X,Y) = \mathbb{E}[k(X_1, X_2)k(Y_1, Y_2)] \qquad (12)$$
$$- 2\mathbb{E}[k(X_1, X_2)k(Y_1, Y_3)] + \mathbb{E}[k(X_1, X_2)]\mathbb{E}[k(Y_1, Y_2)].$$

The real issue is how to estimate (kernel) squared distance covariances. The original approach dates back to Székely & Rizzo (2014), which estimates the original Euclidean distance covariance where $N = m + n$ is the pooled sample size, and is given as:

$$\mathcal{V}_{m,n} = \frac{1}{N(N-3)} \sum_{i \ne j} (\tilde{A}_{ij} \tilde{B}_{ij}), \qquad (13)$$

where $\tilde{A}_{st}$ and $\tilde{B}_{st}$ are $\mathcal{U}$-centered statistics whose definitions can be found in Section 3 of Székely & Rizzo (2014).

Another interesting approach of estimators for (kernel) squared distance covariance is given in Section 4 of Chakraborty & Zhang (2021), which involves both $\text{cdCov}_{m,n}^2(X,Y)$, the cross-distance covariance between $X$ and $Y$, and $\mathcal{V}_m^{k*}$ and $\mathcal{V}_n^{k*}$, where the latter are estimators of centered squared distance variances. Their studentization does not require the pooled sample size to go to infinity; however, they had some extra assumptions which are not relevant to our context. In this article, our studentization adopts the original style of (13), which has also been successfully used in other settings, e.g. in Gao et al. (2021) and Gao & Shao (2023).

#### 5.1.1. KSD COVARIANCE IN SPHERICAL HARMONICS

Now we are ready to apply the KSD covariance estimator in the same style of (13). The $p$-th finite-level kernel squared distance covariance for $H_p = \bigoplus_{i=0}^{2p} \mathcal{H}_i$ is given by:

$$V_p^2(X,Y) := \mathbb{E}[K_p(X_1, X_2)K_p(Y_1, Y_2)] \qquad (14)$$
$$- 2\mathbb{E}[K_p(X_1, X_2)K_p(Y_1, Y_3)] + \mathbb{E}[K_p(X_1, X_2)]\mathbb{E}[K_p(Y_1, Y_2)],$$

where $\{X_i\}_{i=1}^3$ are i.i.d. copies of $X$ and $\{Y_i\}_{i=1}^3$ are i.i.d. copies of $Y$. Under $H_0$, when the two samples $\{X_i\}_{i=1}^m$ and $\{Y_j\}_{j=1}^n$ are obtained from the same distribution $X \overset{d}{=} Y \overset{d}{=} Z$, we can combine these two samples into a pooled sample $\{Z_\ell\}_{\ell=1}^N$, where $N = m + n$ is the pooled sample size, as follows:

$$Z_\ell = \begin{cases} X_\ell, & 1 \le \ell \le m; \\ Y_{\ell-m}, & m < \ell \le N \end{cases} \qquad (15)$$

With this new combined sample $\{Z_\ell\}_{\ell=1}^N$ (15), the KSD covariance becomes $p$-th kernel squared distance variance:

$$V_p^2(Z) := \mathbb{E}[K_p^2(Z_1, Z_2)] - 2\mathbb{E}[K_p(Z_1, Z_2)K_p(Z_1, Z_3)]$$
$$+ \left\{\mathbb{E}[K_p(Z_1, Z_2)]\right\}^2, \qquad (16)$$

where $\{Z_i\}_{i=1}^3$ are i.i.d. copies of $Z$. We can estimate the kernel squared distance covariance with $\{Z_\ell\}_{\ell=1}^N$ under $H_0$. We set up the following notation for every pair of data points using the reproducing kernel in $H_p = \bigoplus_{i=0}^{2p} \mathcal{H}_i$, for $1 \le s \ne t \le N$:

$$_p a_{st}^{m,n} = \sum_{i=0}^{2p} k_i(Z_s, Z_t) = K_p(Z_s, Z_t). \qquad (17)$$

The standard $\mathcal{U}$-centered $_p a_{s,t}^{m,n}$, $s \ne t$, is given as the following:

$$_p A_{st}^{m,n} = {}_p a_{st}^{m,n} - {}_p\tilde{a}_{\cdot t} - {}_p\tilde{a}_{s\cdot} + {}_p\tilde{a}_{\cdot\cdot}$$
$$= {}_p a_{st}^{m,n} - \frac{1}{N-2}\sum_{k=1}^N {}_p a_{kt}^{m,n} - \frac{1}{N-2}\sum_{\ell=1}^N {}_p a_{s\ell}^{m,n}$$
$$+ \frac{1}{(N-1)(N-2)}\sum_{k=1}^N\sum_{\ell=1}^N {}_p a_{k\ell}^{m,n}. \qquad (18)$$

Using $_p A_{st}^{m,n}$ from (18), the $p$-th kernel squared distance covariance is defined as:

$$\mathcal{V}_{m,n}^p := \frac{1}{N(N-3)} \sum_{1 \le s \ne t \le N} \left({}_p A_{st}^{m,n}\right)^2. \qquad (19)$$

## 5.2. The Variance Component via $U$-Statistics Theory

The exact connection between $\mathcal{V}_{m,n}^p$ and $\mathrm{Var}[\hat{\mathcal{E}}_{m,n}^p(X, Y)]$ is very subtle, which is why $U$-statistics theory comes into play in our work. In recent years, $U$-statistics theory has been revitalized as a tool for decomposing estimators and studying their asymptotic properties (Pan et al., 2018; 2019; 2020). Using U-statistics theory to understand the variance of MMD/Energy-type of estimator was pioneered by Huang and Huo in Huang & Huo (2017), and was later simplified and greatly generalized in later works, e.g., (Gao & Shao, 2023; Chakraborty & Zhang, 2021; Gao et al., 2021).

All those works in Gao & Shao (2023), Chakraborty & Zhang (2021) and Yan & Zhang (2023) essentially lead to equivalent asymptotic results in Euclidean contexts. Among those works, the most adaptable approach to our non-Euclidean context is the one in Gao & Shao (2023), however, we cannot directly apply their intermediate results in our spherical/compositional context, not only for the apparent reasons (listed at the end of Section 4.3), but also because our kernels are not characteristic as in Gao & Shao (2023). Therefore, we must carefully examine which of the intermediate results in Gao & Shao (2023) can be "translated" to our setting, and which need to be re-established, for the purposes of studentization in our context.

### 5.2.1. Variance Component in Terms of $\mathcal{V}_{m,n}^p$ under Null Hypotheses

The purpose of this subsection is to re-develop some key intermediate results parallel to those in Gao & Shao (2023) in our non-Euclidean contexts for compositional/directional data. However, the core of the argument is still based on the Hoeffding decomposition in $U$-statistics theory.

In order to relate the estimator $\mathcal{E}_{m,n}^p$ in (11) with $U$-statistics theory, we need to rewrite it in terms of two sample kernels. For our purposes, the $p$-th level two-sample kernel can be simply defined as:

$$h_p(X_1, X_2, Y_1, Y_2) := K_p(X_1, X_2) + K_p(Y_1, Y_2)$$
$$- \frac{1}{2}\sum_{i=1}^2\sum_{j=1}^2 K_p(X_i, Y_j). \qquad (20)$$

This finite $p$-th level two sample kernel in (20) will be the $U$-statistics kernel for the following discussions. The estimator $\mathcal{E}_{m,n}^p$ defined in (11) can be rewritten in terms of the finite-level two-sample kernel (20) as

$$\mathcal{E}_{m,n}^p(X, Y)$$
$$= \binom{m}{2}^{-1}\binom{n}{2}^{-1} \sum_{\substack{1 \le i_1 < i_2 \le m \\ 1 \le j_1 < j_2 \le n}} h_p(X_{i_1}, X_{i_2}, Y_{j_1}, Y_{j_2}).$$

As is well-known from U-statistics theory (see Section 2 in Hoeffding (1948)), the kernel-based statistic $\mathcal{E}_{m,n}^p$ admits a Hoeffding decomposition:

$$\mathcal{E}_{m,n}^p = \sum_{a=0}^2\sum_{b=0}^2 \binom{2}{a}\binom{2}{b} {}_p H_{m,n}^{(a,b)}, \qquad (21)$$

where each Hoeffding component is defined in (26) in Appendix A.2. Let $_p\mathcal{H}_{m,n}^{(a,b)} = \binom{2}{a}\binom{2}{b} {}_p H_{m,n}^{(a,b)}$, then the decomposition in (21) is just $\mathcal{E}_{m,n}^p = \sum_{a=0}^2\sum_{b=0}^2 {}_p\mathcal{H}_{m,n}^{(a,b)}$, which can be further decomposed as:

$$\mathcal{E}_{m,n}^p = \underbrace{\sum_{a+b\le 2} {}_p\mathcal{H}_{m,n}^{(a,b)}}_{L_{m,n}^p} + \underbrace{\sum_{a+b>2} {}_p\mathcal{H}_{m,n}^{(a,b)}}_{R_{m,n}^p}. \qquad (22)$$

Decomposition (22) further decomposes $\mathcal{E}_{m,n}^p$ into a "linear term" $L_{m,n}^p$ and a "remainder term" $R_{m,n}^p$. This decomposition was also used in Gao & Shao (2023), although their context was about infinite-dimensional RKHS on $\mathbb{R}^p$. However, even for our finite dimensional RKHS $\bigoplus_{i=0}^{2p} \mathcal{H}_i$ for a fixed underlying $\mathbb{S}^d$, we can still obtain the following analogous result:

**Proposition 5.1.** *The remainder term $R_{m,n}^p$ in decomposition (22) vanishes for every positive integer $p$, i.e., $R_{m,n}^p = 0$.*

**Proof:** Details are given in Appendix A.3.1. □

With Proposition 5.1 simplifying $\mathcal{E}_{m,n}^p$ to the linear component $L_{m,n}^p$, the next step is to relate $\text{Var}(\mathcal{E}_{m,n}^p) = \text{Var}(L_{m,n}^p)$ to the KSD covariances and their estimators.

Under $H_0$, where $X \overset{d}{=} Y \overset{d}{=} Z$, the pooled sample $\{Z_\ell\}_{\ell=1}^N$ has the $p$-th KSD variance (16) as follows:

$$V_p^2(Z) = \mathbb{E}[K_p^2(Z_1, Z_2)] - 2\mathbb{E}[K_p(Z_1, Z_2)K_p(Z_1, Z_3)] + \left\{\mathbb{E}[K_p(Z_1, Z_2)]\right\}^2.$$

In our context, we can still relate the variance component $\text{Var}(\mathcal{E}_{m,n}^p) = \text{Var}(L_{m,n}^p)$ to the KSD variance $V_p^2(Z)$ through the following proposition (recovering an analogous property in Gao & Shao (2023)):

**Proposition 5.2.** *Under $H_0$, $X \overset{d}{=} Y \overset{d}{=} Z$, for two independent i.i.d. samples $\{X_i\}_{i=1}^m$ and $\{Y_j\}_{j=1}^n$, for every positive integer $p$, we have*

$$\text{Var}(\mathcal{E}_{m,n}^p) = \text{Var}(L_{m,n}^p)$$
$$= \underbrace{\left[\binom{m}{2}^{-1} + \binom{n}{2}^{-1} + \frac{4}{mn}\right]}_{c_{m,n}} V_p^2(Z) = c_{m,n} V_p^2(Z).$$

Based on the resemblance between our Proposition 5.2 and a proposition in (Gao & Shao, 2023), one would hope to directly apply their result in our setting. However, their setting involves infinitely many *different* underlying Euclidean spaces, whereas ours involves infinitely many RKHSs over *the same* underlying *non-Euclidean* space. Therefore, we have to re-derive a parallel version of the proposition in our context. The complete proof is given in Appendix A.3.

### 5.3. Studentization of Two Sample Statistics

So far, we have only established the connection between $\text{Var}(\mathcal{E}_{m,n}^p) = \text{Var}(L_{m,n}^p)$ and the theoretical quantity $V_p^2(Z)$ in Proposition 5.2, but to construct test statistics, we need to replace $V_p^2(Z)$ with its estimator. Recall that the estimator for $V_p^2(Z)$ is given by $\mathcal{V}_{m,n}^p$ in (19), and we need a consistency result for this estimator $\mathcal{V}_{m,n}^p$.

**Lemma 5.3.** *Let the pooled sample size $N = m + n$ go to infinity for the sphere dimension $d$ fixed with the following additional requirements:*

*1. Growth rate condition relative to "$p$":*

$$\lim_{N\to\infty} \frac{p^d}{N} = 0;$$

*2. Asymptotic ratio consistency between the two samples:*

$$\lim_{N\to\infty} \frac{m}{N} = \lim_{N\to\infty}\left(1 - \frac{n}{N}\right) = \rho, \quad 0 < \rho < 1.$$

*Then, under $H_0 : X \overset{d}{=} Y \overset{d}{=} Z$, we have:*

$$\frac{\mathcal{V}_{m,n}^p}{V_p^2(Z)} \longrightarrow 1, \text{ in probability as } N, p \to \infty.$$

*Remark* 5.4. The first condition in Lemma 5.3 requires $p^d/N \to 0$ as $N \to \infty$, which may suggest that the required sample size might be too large, but the key is that $p$ diverges much more slowly than $N$, so one can simply choose $p = \lfloor (\log N)^\delta \rfloor$ for some $\delta > 0$ in practice.

Plugging Proposition 5.2 into Lemma 5.3, we obtain the following convergence (by Proposition 5.2):

$$\frac{\mathcal{V}_{m,n}^p}{V_p^2(Z)} = \frac{c_{m,n}\mathcal{V}_{m,n}^p}{c_{m,n}V_p^2(Z)} = \frac{c_{m,n}\mathcal{V}_{m,n}^p}{\text{Var}(\mathcal{E}_{m,n}^p)} \overset{p}{\to} 1.$$

Hence, the natural estimator of the variance component $\text{Var}(\mathcal{E}_{m,n}^p)$ is "$c_{m,n}\mathcal{V}_{m,n}^p$". Therefore, we can define our studentized test statistic:

$$T_{m,n}^p = \frac{\mathcal{E}_{m,n}^p}{\sqrt{c_{m,n}\mathcal{V}_{m,n}^p}}. \tag{23}$$

This test statistic in (23) looks rather similar to the one in Gao & Shao (2023), but again their test statistic $T_{m,n,p}^k$ was built *on* a sequence of vector spaces $\{\mathbb{R}^p\}_{p\to\infty}$, while our statistic $T_{m,n}^p$ is based on a sequence of reproducing kernel Hilbert spaces $\{\bigoplus_{i=0}^{2p} \mathcal{H}_i\}_{p\to\infty}$ for the fixed underlying sphere $\mathbb{S}^d$; nonetheless, we still obtain a similar asymptotic result for the asymptotic null distribution:

**Theorem 5.5.** *Under $H_0$, $X \overset{d}{=} Y \overset{d}{=} Z$, the same growth rate condition $p^d/N \to 0$ and the asymptotic ratio condition $m/N \to \rho$ as in Lemma 5.3, in addition to the requirement that probability density functions are continuous on spherical/compositional domains[5], the test statistic $T_{m,n}^p$ in (23) has the following asymptotic distribution:*

$$T_{m,n}^p \overset{d}{\longrightarrow} N(0,1), \text{ as } N \text{ and } p \to \infty.$$

The statement of Theorem 5.5 is similar to Theorem 16 in Gao & Shao (2023), *but* without its complicated assumptions on the asymptotic growth of reproducing kernels. Indeed, our sequence of reproducing kernels $\{K_p\}_{p\geq 1}$ is *a priori* given, so we *cannot* assume their asymptotic conditions for kernels, but instead we have to *prove* that our sequence of kernels *satisfies* those asymptotic growth conditions. Therefore, demonstrating those asymptotic conditions on our sequence of kernels will be the key component in our proof of Theorem 5.5, which is given in Appendix A.4.

---

[5]Theorem 5.5 considered only continuous distributions on spherical/compositional domains, however, the arguments also work for bounded discrete distributions. We also believe that this theorem should be true even for distributions with poles.

# 6. Experiments

We have developed both compositional and directional non-parametric two-sample test. We present simulation results for both data types and compare with existing approaches. Appendix B provides details of the simulation studies, while Appendix B.6.2 presents a comparison study on real compositional data.

## 6.1. Compositional Two Sample Simulation Study

In our experiment, since $\mathrm{KED}_p(P,Q) \geq 0$, larger values indicate stronger evidence against $H_0$. Thus, at level $\alpha$, we reject $H_0 : P = Q$ when $T_{m,n}^p > z_{1-\alpha}$, the $(1-\alpha)$ quantile of the asymptotic standard normal distribution $N(0,1)$.

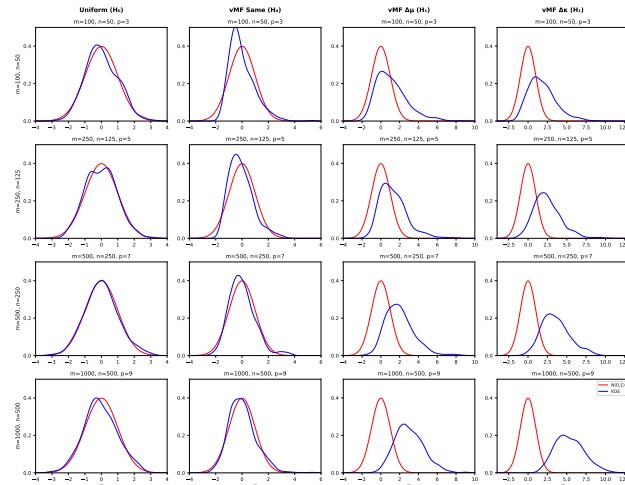

*Figure 3.* KDE of the studentized test statistic (blue) overlaid with the $N(0,1)$ distribution (red). The first two columns show convergence to $N(0,1)$ under $H_0$ as sample size increases. For the alternatives in columns 3 and 4, we observe increasing distance from the null distribution as the sample size increases from the top row ($m = 100, n = 50$) to the bottom row ($m = 1000, n = 500$). Traversing down a column represents increasing sample size.

| Test Statistic | Null | $\mu$: rare | $\mu$: abundant | $\mu$: uniform | concentration |
|---|---|---|---|---|---|
| CLR-ED | 0.03 | 0.38 | 0.05 | 0.57 | 0.33 |
| CLR-MMD | 0.03 | 0.39 | 0.05 | 0.55 | 0.31 |
| SPH-$\sqrt{}$-p2 | 0.07 | 0.46 | 1.00 | 1.00 | 0.35 |
| SPH-$\sqrt{}$-p4 | 0.08 | 0.46 | 1.00 | 1.00 | 0.83 |
| SPH-L2-p2 | 0.07 | 0.09 | 1.00 | 1.00 | 0.60 |
| SPH-L2-p4 | 0.07 | 0.06 | 1.00 | 1.00 | 0.93 |

*Table 1.* Empirical rejection rates for compositional two-sample tests under Dirichlet–Multinomial data ($\alpha = 0.05$, $d = 10$, $m = n = 100$, both samples containing approximately 5% zeros, CLR pseudocount $\varepsilon = 10^{-8}$). *Null*: $P = Q$ (target 0.05); $\mu$: *rare/abundant/uniform*: mean shift $\delta = 0.25$ on rare, abundant, or all components; *concentration*: matched means with Dirichlet concentration scaled by $\eta = 2.0$.

In Table 1, we compared our test statistic, the methods SPH-$\sqrt{}$-p2, SPH-$\sqrt{}$-p4, SPH-L2-p2, SPH-L2-p4, to Centered Log Ratio (CLR) based methods via Dirichlet-Multinomial simulations under mean shift and concentration alternatives. CLR performed comparably when differences occurred in rare taxa, but had near zero power (5%) for abundant taxa, where our method achieved 100% power. For concentration alternatives, SPH-L2-p4 substantially outperformed CLR (93% vs 33%). Full details and definitions are in the appendix B.3.

## 6.2. Directional Two Sample Simulation Study

We simulated four cases of spherical data shown in Figure 5, with $d = 2$, imbalanced samples ratio $\frac{m}{n} = 2$, and 500 replications per scenario. **Results: (1)** Convergence to $N(0,1)$ under $H_0$ with uniform and von Mises-Fisher distributions; **(2)** For the alternative scenario in column 3 we sample from von Mises-Fisher distributions with means differing by $7°$; **(3)** In the fourth column alternative scenario, means are identical, but the ratio of concentration parameters is $\frac{\kappa_X}{\kappa_Y} = 1.75$. Our test statistic is able to distinguish distributions on the sphere even **when the means are identical, but higher order moments differ**.

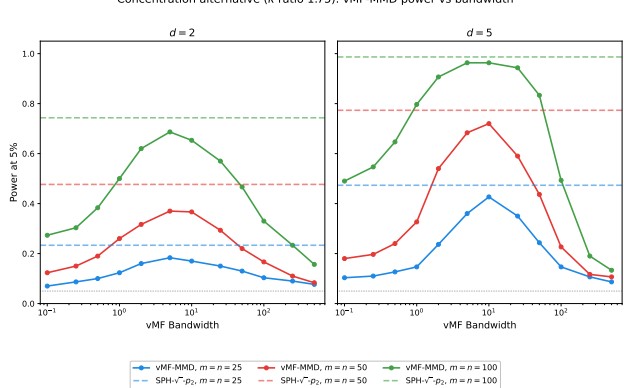

*Figure 4.* Power at $\alpha = 0.05$ versus kernel bandwidth ($d = 2$ left, $d = 5$ right; vMF concentration alternative, $\kappa$ ratio 1.75). Solid: vMF-MMD, $m = n \in \{25, 50, 100\}$ (blue/red/green). Dashed: proposed SPH-$\sqrt{}$-$p_2$ (bandwidth-free), same sample sizes. For example, to interpret these results, note that the green curves correspond to the indicated sample size regime, the green dashed line is the (bandwidth independent) performance of our method, the solid green curve is the performance of the vMF-MMD as a function of the bandwidth parameter, etc.

A natural choice of a competing test statistic within the domain of directional data, is the vMF-MMD kernel. The vMF-MMD requires the choice of a bandwidth parameter, and its power is bell-shaped in the bandwidth. Of course, in practice the optimal bandwidth value is not knowable. Figure 4 shows that our proposed SPH-$\sqrt{}$-$p_2$ test statistic has higher power than vMF-MMD at every bandwidth and sample size. Regardless, in addition to superior performance,

our method does not require any such bandwidth parameter selection.

## 7. Discussion

Our nonparametric two-sample test for compositional and directional data via studentized spherical harmonics-based test statistics does enjoy nice asymptotic normality and accurate size under $H_0$, and achieves higher empirical power under $H_1$ compared to log-transform-based MMD methods for CoDA. Even when directional data have different distributions but the same mean direction, our method still effectively detected distributional differences.

However, the theoretical regime of this paper is developed under a fixed-dimensional sphere with dimension $d$, so the sample size can be controlled only for this fixed dimension to satisfy the first condition in Lemma 5.3; even if we choose $p = 2$, the minimum required sample size exceeds $2^d$, so when the dimension is very high, it is still necessary to develop a separate theoretical framework for high-dimensional low sample size (HDLSS) two-sample tests for compositional and directional data.

Moreover, the HDLSS two-sample test will also be non-Euclidean in nature, and extra care needs to be taken in addressing compatibility between lower- and higher-dimensional spheres, because the underlying spaces will vary. The current work deals with different function spaces over the same underlying sphere; however, it is another subtle issue of how to handle function spaces from *different underlying spaces* in a coherent way in order to derive meaningful asymptotic results. The authors will address these issues in forthcoming work.

## Software and Data

Implementation code, simulation code, and data are available at GitHub.[6]

## Acknowledgments

Kim is partially supported by NSF DMS-EiR Grant #2100729. Binglin thanks Lynn Baragona for some financial assistance that supported his travel during this work. Matthew thanks Deep AI, Inc. for their financial support during this work.

## Impact Statement

This paper not only develops the first completely nonparametric two sample framework for both compositional

data analysis and directional statistics *without* any dimensional constraints, but also, more importantly, initiates a potentially new paradigm for Non-Euclidean data analysis by translating a fixed-dimensional non-Euclidean data question to a statistics problem in the realm of classical asymptotic high-dimensional data analysis. Unlike traditional Non-Euclidean manifold data analysis, which heavily relies on local charts, this new approach focuses on linearizing global aspects of Non-Euclidean spaces and breaks down to problems in the more familiar domain of classical high-dimensional framework.

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

# A. Appendix

In this section, we provide supplementary material for our main article, but also sketch key idea and important strategies for the proofs of mathematical results from various sections.

## A.1. Review of Spherical Harmonics

In Section 3.1, it concluded that instead of using compositional kernels directly, we go back to the spread-out philosophy to the whole sphere and directly use spherical kernels on this spread-out data set. The spherical kernels are reproducing kernels for $\mathcal{H}_i$, the fundamental building block of spherical harmonics theory. The following proposition reveals the relation of spherical kernel in $\mathcal{H}_i$ and Gegenbauer polynomials:

**Proposition A.1.** *For each spherical harmonics subspace $\mathcal{H}_i \subset L^2(\mathbb{S}^d)$, there exists a reproducing kernel $k_i(\cdot, \cdot)$:*

$$k_i(\cdot, \cdot) : \ \mathbb{S}^d \times \mathbb{S}^d \to \mathbb{R},$$

*such that the followings hold:*

(i) *For any $f \in \mathcal{H}_i$, we have the following reproducing property:*

$$< f(t), k_i(x, t) >_{\mathbb{S}^d} = \int_{\mathbb{S}^d} f(t) k_i(x, t) dt = f(x), \ \forall x \in \mathbb{S}^d.$$

(ii) *The spherical kernel $k_i(x, t)$ can be "homogenized" to a homogeneous polynomial in both $x$ and $t$ of degree $i$, namely $k_i(x, t)$ is a degree $i$ homogeneous polynomial in $t$ for every fixed $x \in \mathbb{S}^d$, and similarly $k_i(x, t)$ is also a degree $i$ homogeneous polynomial in $x$ for every $t \in \mathbb{S}^d$.*

(iii) *The kernel $k_i(x, t)$ is orthogonal invariant, and is completely determined by the Gegenbauer polynomial $C_i^\lambda(t)$, namely,*

$$k_i(x, t) = h_{i,d} C_i^\lambda(< x, t >),$$

*where $\lambda = (d-1)/2$ and $h_{i,d} = a_{i,d}/[\text{vol}(\mathbb{S}^d) C_i^\lambda(1)]$ with $a_{i,d} = \binom{d+i}{d} - \binom{d+i-2}{d}$.*

**Proof:** These results are well known in spherical harmonics theory, which can be found in the 2nd Section of Chapter IV from (Stein & Weiss, 1971) by Elias Stein and Guido Weiss. □

In this article, we will need to consider infinite direct sums $\bigoplus_{i=0}^\infty \mathcal{H}_i$ of all spherical harmonic subspaces, but finite direct sums of those subspaces have close relations with homogeneous polynomials.

Let $\mathcal{P}_i(d+1)$ be the space of degree $i$ homogeneous polynomials on $(d+1)$ variables, then the connection between spherical harmonics and homogeneous polynomials can be revealed as follows:

$$\begin{cases} \bigoplus_{i=0}^n \mathcal{H}_{2i} & \cong \quad \mathcal{P}_{2n}(d+1); \\ \\ \bigoplus_{i=0}^n \mathcal{H}_{2i+1} & \cong \quad \mathcal{P}_{2n+1}(d+1). \end{cases} \tag{24}$$

It might seem odd for beginners to see two different cases of direct sums in (24), but it is intuitive from classical Fourier expansion theory:

- A priori, a function in the direct sum $\bigoplus_{i=0}^p \mathcal{H}_{2i}$ (or $\bigoplus_{i=0}^n \mathcal{H}_{2i+1}$) is not necessarily given in a homogeneous form, but one can apply a "homogenization" process to get a homogeneous polynomial.
- Classical Fourier series are expanded by the bases $\{\sin(kx)\}_{k \in \mathbb{Z}_{>0}}$ and $\{\cos(kx)\}_{k \in \mathbb{Z}_{\geq 0}}$; spherical harmonics is a generalization of classical Fourier expansions for higher dimensional spheres.

In this article, essentially we want to consider the direct sum of all spherical harmonics $\lim_{p \to \infty} \bigoplus_{i=0}^p \mathcal{H}_i$; our strategy is to take the finite level direct sum $\bigoplus_{i=0}^{2p} \mathcal{H}_i$, and build our test statistics out of this finite level direct sum, before taking the limit $p \to \infty$.

The reason we chose this finite level sum $\bigoplus_{i=0}^{2p} \mathcal{H}_i$ for every $p$ is mainly out of convenience, because many properties can be uniformly discussed independent of the parity of the integer $p$: for example the dimension of $\bigoplus_{i=0}^{2p} \mathcal{H}_i$ can be simply expressed as $\binom{d+2p}{d} + \binom{d+2p-1}{d}$, no matter $p$ is even or odd; as a consequence, many arguments for the proof of our major results will be much more streamlined.

### A.2. All Necessary Notions of $U$-statistics in This Article

The following quantity is the key component of Hoeffding decomposition for $\mathcal{E}_{m,n}^p$ (with $0 \leq a, b \leq 2$):

$$
\begin{aligned}
& h_p^{(c,d)}(x_1, \cdots, x_c; y_1, \cdots, y_d) \\
&= \int h_p(u_1, u_2, v_1, v_2) \prod_{i=1}^{c} [d\delta_{x_i}(u_i) - dF_X(u_i)] \prod_{i=c+1}^{2} dF_X(u_i) \prod_{j=1}^{d} [d\delta_j(v_j) - dF_Y(v_j)] \prod_{j=d+1}^{2} dF_Y(v_j).
\end{aligned} \tag{25}
$$

Each Hoeffding component in (21) is the following for $0 \leq a, b \leq 2$:

$$
\begin{aligned}
& {}_pH_{m,n}^{(a,b)} \\
&= \binom{m}{a}^{-1} \binom{n}{b}^{-1} \sum_{(m,a)} \sum_{(n,b)} h_p^{(a,b)}(X_{i_1}, \ldots, X_{i_a}, Y_{j_1}, \ldots, Y_{j_b});
\end{aligned} \tag{26}
$$

here the symbol $(m, a)$ that underlies the summation symbol for ${}_pH_{m,n}^{(a,b)}$ refers to a choice of $a$ observations from the sample $\{X_i\}_{i=1}^m$ for the first $a$ arguments in $h_p^{(a,b)}$ (in (25)) and similarly $(n, b)$ refers to a choice of $b$ observations from the sample $\{Y_j\}_{j=1}^n$ for the second $b$ arguments in $h_p^{(a,b)}$, while the summation $\sum_{(m,a)}$ (resp. $\sum_{(n,b)}$) takes the sum over all choices of $a$ observations from the sample $\{X_i\}_{i=1}^m$ (resp. $b$ observations of the sample $\{Y_j\}_{j=1}^n$) for the function $h_p^{(a,b)}$.

### A.3. Proof of Proposition 5.2

Our original hope is to directly apply Proposition 9 in (Gao & Shao, 2023) to achieve the result in Proposition 5.2, but unfortunately our context is different, and we need to write a parallel version of their proof in our context of spherical harmonics. Due to the technical nature of this proof, we will try to streamline the exposition of this proof as well.

As was remarked at the end of Section 5.2.1, Proposition 5.2 is not just a quantitative result, and its key breaks down to the existence of adaptive bases which are used to construct certain type of bi-varate functions related to $h_{2,0}^p$, $h_{0,2}^p$ and $h_{1,1}^p$ (defined in (28)) via Mercer's theorem, via Mercer's theorem, before leveraging Martingale difference sequences to compute the variance component.

### A.3.1. BREAKING $h_p^{(c,d)}$ INTO SUMS OF $h_{a,b}^p$'S AND THE PROOF OF PROPOSITION 5.1

Recall that $h_p^{(c,d)}$ was defined in (25), which are fundamental pieces in the Hoeffding decomposition. Specified to the H-decomposition of our estimator $\mathcal{E}_{m,n}^p$ in (22), our goal is to show that the residual term $R_{m,n}^p = 0$. Note that the residual term $R_{m,n}^p$ is spanned by $h_p^{(c,d)}$'s with $c + d > 2$, hence it suffices for us to prove that $h_p^{(2,1)} = h_p^{(1,2)} = h_p^{(2,2)} = 0$; more specifically if we plug in the random variables from two samples, our proof follows if we can show the following equalities:

$$
\begin{cases}
h_p^{(2,1)}(X_1, X_2, Y_1) &= 0, \\
h_p^{(1,2)}(X_1, Y_1, Y_2) &= 0, \\
h_p^{(2,2)}(X_1, X_2, Y_1, Y_2) &= 0,
\end{cases} \tag{27}
$$

where $\{X_1, X_2\}$ are independent copies of $X$ and $\{Y_1, Y_2\}$ are independent copies of $Y$.

The following term plays an important role to to understand $h_p^{(c,d)}$:

$$
\begin{aligned}
h_{a,b}^p &= h_{a,b}^p(X_1, \ldots, X_a, Y_1, \ldots, Y_b) \\
&= \mathbb{E}_{\{\{X_i\}_{i>a}, \{Y_j\}_{j>b}\}}[h_p(X_1, X_2, Y_1, Y_2)].
\end{aligned} \tag{28}
$$

In practice, it is much easier to work with $h_{c,d}^p$ in (25), because it is directly defined in terms of expectations from two sample kernels. The following lemma establishes the relation of $h_p^{(c,d)}$ with $h_{c,d}^p$'s:

**Lemma A.2.** *For every positive integer $p$, the following equality holds:*

$$h_p^{(c,d)} = \sum_{0 \le a \le c} \sum_{0 \le b \le d} (-1)^{(c+d)-(a+b)} \sum_{(2,a)} \sum_{(2,b)} h_{a,b}^p$$

The form of Lemma A.2 is summarized in a more general form than the one in (Gao & Shao, 2023), and we also used the same notational convention as in (26). The proof can simply break down by following the definition of $h_p^{(c,d)}$ through the integration given in (25) for every pair of $(c,d)$ where $c, d \ge 0$ with $c + d > 2$.

**Corollary A.3.** *The equations in (27) all hold, namely $h_p^{(2,1)} = h_p^{(1,2)} = h_p^{(2,2)} = 0$, hence in particular $R_{m,n}^p = 0$.*

*Proof.* We will explain the basic idea in here: recall that $h_{a,b}^p$ was defined through the expectation of the two sample kernel in (20):

$$h_p(X_1, X_2, Y_1, Y_2) = K_p(X_1, X_2) + K_p(Y_1, Y_2) - \frac{1}{2} \sum_{1 \le i,j \le 2} K_p(X_i, Y_j)$$

via the (partial) expectation $h_{a,b}^p = \mathbb{E}_{\{\{X_i\}_{i>a}, \{Y_j\}_{j>b}\}}[h_p(X_1, X_2, Y_1, Y_2)]$, therefore we can express $h_{a,b}^p$ in terms of (partial) expectations of reproducing kernel. Note that $\{X_1, X_2\}$ are independent copies of $X$ and $\{Y_1, Y_2\}$ are independent copies of $Y$, so we can simply the expression of $h_{a,b}^p$ by combining a few terms e.g. $\mathbb{E}_{Y_1}[K_p(X_1, Y_1)] + \mathbb{E}_{Y_2}[K_p(X_1, Y_2)] = 2\mathbb{E}_Y[K_p(X_1, Y)]$. By using this simplication strategy on $h_{a,b}^p$ before plugging in Lemma A.2, this corollary easily follows.

We conclude this exposition of this proof by demonstrating one concrete example by showing that $h_p^{(1,2)}(X_1, Y_1, Y_2) = 0$, while other cases of $h_p^{(c,d)}$ with $c + d > 2$ follow from the same exact strategy as this example.

From Lemma A.2, we know that $h_p^{(1,2)}$ is an alternating sum of $h_{1,2}^p$, $h_{1,1}^p$, $h_{1,0}^p$, $h_{0,2}^p$, $h_{0,1}^p$ and $h_{0,0}^p$. We can compute each of them $h_{a,b}^p$ (with $a \le 1$ and $b \le 2$) applying the (partial) expectations of two sample kernels and obtain the following:

$$
\begin{cases}
h_{0,0}^p & = \ \mathbb{E}[K_p(X_1, X_2)] + \mathbb{E}[K_p(Y_1, Y_2)] - 2\mathbb{E}[K_p(X, Y)]; \\[2mm]
h_{1,0}^p(X_1) & = \ \mathbb{E}_X[K_p(X_1, X)] + \mathbb{E}[K_p(Y_1, Y_2)] - \mathbb{E}_Y[K_p(X_1, Y)] - \mathbb{E}[K_p(X, Y)]; \\[2mm]
h_{1,0}^p(X_2) & = \ \mathbb{E}_X[K_p(X, X_2)] + \mathbb{E}[K_p(Y_1, Y_2)] - \mathbb{E}_Y[K_p(X_2, Y)] - \mathbb{E}[K_p(X, Y)]; \\[2mm]
h_{0,1}^p(Y_1) & = \ \mathbb{E}[K_p(X_1, X_2)] + \mathbb{E}_Y[K_p(Y_1, Y)] - \mathbb{E}_X[K_p(X, Y_1)] - \mathbb{E}[K_p(X, Y)]; \\[2mm]
h_{0,1}^p(Y_2) & = \ \mathbb{E}[K_p(X_1, X_2)] + \mathbb{E}_Y[K_p(Y, Y_2)] - \mathbb{E}_X[K_p(X, Y_2)] - \mathbb{E}[K_p(X, Y)]; \\[2mm]
h_{2,0}^p(X_1, X_2) & = \ K_p(X_1, X_2) + \mathbb{E}[K_p(Y_1, Y_2)] - \mathbb{E}_Y[K_p(X_1, Y)] - \mathbb{E}_Y[K_p(X_2, Y)]; \\[2mm]
h_{0,2}^p(Y_1, Y_2) & = \ \mathbb{E}[K_p(X_1, X_2)] + K_p(Y_1, Y_2) - \mathbb{E}_X[K_p(X, Y_1)] - \mathbb{E}_X[K_p(X, Y_2)]; \\[2mm]
h_{1,1}^p(X_1, Y_1) & = \ \mathbb{E}_X[K_p(X_1, X)] + \mathbb{E}_Y[K_p(Y_1, Y)] \\
& \quad - \frac{1}{2}K_p(X_1, Y_1) - \frac{1}{2}\mathbb{E}_Y[k_p(X_1, Y)] - \frac{1}{2}\mathbb{E}_X[K_p(X, Y_1)] - \frac{1}{2}\mathbb{E}[K_p(X, Y)]; \\[2mm]
h_{1,1}^p(X_1, Y_2) & = \ \mathbb{E}_X[K_p(X_1, X)] + \mathbb{E}_Y[K_p(Y, Y_2)] \\
& \quad - \frac{1}{2}\mathbb{E}_Y[k_p(X_1, Y)] - \frac{1}{2}K_p(X_1, Y_2) - \frac{1}{2}\mathbb{E}[K_p(X, Y)] - \frac{1}{2}\mathbb{E}_X[K_p(X, Y_2)]; \\[2mm]
h_{1,2}^p(X_1, Y_1, Y_2) & = \ \mathbb{E}_X[K_p(X_1, X)] + K_p(Y_1, Y_2) - \frac{1}{2}\sum_{i=1}^{2}\mathbb{E}_X[K_p(X, Y_i)] - \frac{1}{2}\sum_{i=1}^{2}k_p(X_1, Y_i)
\end{cases}
\tag{29}
$$

According to Lemma A.2, we can decompose $h_p^{(1,2)}(X_1, Y_1, Y_2)$ as follows:

$$
\begin{aligned}
h_p^{(1,2)}(X_1, Y_1, Y_2) = \ & h_{1,2}^p(X_1, Y_1, Y_2) - h_{0,2}^p(Y_1, Y_2) - [h_{1,1}^p(X_1, Y_1) + h_{1,1}^p(X_1, Y_2)] \\
& + h_{1,0}^p(X_1) + [h_{0,1}^p(Y_1) + h_{0,1}^p(Y_2)] - h_{0,0}^p.
\end{aligned}
\tag{30}
$$

Plugging (29) into (30), it is a simple arithmetic computation to conclude that $h_p^{(1,2)}(X_1, Y_1, Y_2) = 0$. $\qquad\square$

### A.3.2. EXISTENCE OF ADAPTABLE BASES AND MERCER'S THEOREM IN SPHERICAL HARMONICS

Following from Proposition 5.1, the Hoeffding decomposition of the estimator $\mathcal{E}_{m,n}^p$ has has the "linear term" remained, namely we have the following:

$$\mathcal{E}_{m,n}^p = L_{m,n}^p = \sum_{c+d\leq 2} {}_p\mathcal{H}_{m,n}^{(a,b)},$$

where ${}_p\mathcal{H}_{m,n}^{(a,b)} = \binom{2}{a}\binom{2}{b} {}_pH_{m,n}^{(a,b)}$ with ${}_pH_{m,n}^{(a,b)} = \binom{m}{a}^{-1}\binom{n}{b}^{-1} \sum_{(m,a)}\sum_{(n,b)} h_p^{(a,b)}(X_{i_1}, \ldots, X_{i_a}, Y_{j_1}, \ldots, Y_{j_b})$, $a+b \leq$ 2. Therefore, if we expand this Hoeffding decomposition on $a+b \leq 2$, we get the following expression:

$$
\begin{aligned}
\mathcal{E}_{m,n}^p &= L_{m,n}^p \\[2mm]
&= \sum_{a+b\leq 2} \binom{2}{a}\binom{2}{b} {}_pH_{m,n}^{(a,b)} \\[2mm]
&= h_p^{(0,0)} + \frac{2}{m}\sum_{i=1}^m h_p^{(1,0)}(X_i) + \frac{2}{n}\sum_{j=1}^n h_p^{(0,1)}(Y_j) + \frac{4}{mn}\sum_{i=1}^m\sum_{j=1}^n h_p^{(1,1)}(X_i, Y_j) \\[2mm]
&\quad + \binom{m}{2}^{-1}\sum_{0\leq i_1<i_2\leq m} h_p^{(2,0)}(X_{i_1}, X_{i_2}) + \binom{n}{2}^{-1}\sum_{0\leq j_1<j_2\leq n} h_p^{(0,2)}(Y_{j_1}, Y_{j_2})
\end{aligned}
\tag{31}
$$

An important principle is to break $h_p^{(a,b)}$ into sums of $h_{a,b}^p$'s, whose formula was completely given in Lemma A.2, and we can write down this decomposition in the following:

$$
\begin{cases}
h_p^{(2,0)}(X_1, X_2) &= h_{2,0}^p(X_1, X_2) - h_{1,0}^p(X_1) - h_{1,0}^p(X_2) + h_{0,0}^p; \\[2mm]
h_p^{(0,2)}(Y_1, Y_2) &= h_{0,2}^p(Y_1, Y_2) - h_{0,1}^p(Y_1) - h_{0,1}^p(Y_2) + h_{0,0}^p; \\[2mm]
h_p^{(1,1)}(X, Y) &= h_{1,1}^p(X, Y) - h_{1,0}^p(X) - h_{0,1}^p(Y) + h_{0,0}^p; \\[2mm]
h_p^{(1,0)}(X) &= h_{1,0}(X) - h_{0,0}^p; \\[2mm]
h_p^{(0,1)}(Y) &= h_{0,1}(Y) - h_{0,0}^p; \\[2mm]
h_p^{(0,0)} &= h_{0,0}^p.
\end{cases}
\tag{32}
$$

If we plug (32) into (31), we can completely express $L_{m,n}^p$ in terms of $h_{a,b}^p$'s, then with a careful calculation, we obtain the following expression:

$$
\begin{aligned}
L_{m,n}^p &= \sum_{a+b\leq 2} \binom{2}{a}\binom{2}{b} {}_pH_{m,n}^{(a,b)} \\[2mm]
&= 3h_{0,0}^p - \frac{4}{m}\sum_{i=1}^m h_{1,0}^p(X_i) - \frac{4}{n}\sum_{j=1}^n h_{0,1}^p(Y_j) + \frac{4}{mn}\sum_{i=1}^m\sum_{j=1}^n h_{1,1}^p(X_i, Y_j) \\[2mm]
&\quad + \binom{m}{2}^{-1}\sum_{0\leq i_1<i_2\leq m} h_{2,0}^p(X_{i_1}, X_{i_2}) + \binom{n}{2}^{-1}\sum_{0\leq j_1<j_2\leq n} h_{0,2}^p(Y_{j_1}, Y_{j_2}).
\end{aligned}
\tag{33}
$$

Recall that Proposition 5.2 was under the null hypothesis $H_0: X \stackrel{d}{=} Y$, then under $H_0$ from (29), we have the following

vanishing of "odd degree terms" (i.e. $a + b$ is odd):

$$
\begin{aligned}
h_{1,0}^p(X_i) &= \mathbb{E}_X[K_p(X_i, X)] + \mathbb{E}[K_p(Y_1, Y_2)] - \mathbb{E}_Y[K_p(X_i, Y)] - \mathbb{E}[K_p(X, Y)] \\
&\overset{H_0}{=} \underbrace{\mathbb{E}_X[K_p(X_i, X)] - \mathbb{E}_Y[K_p(X_i, Y)]}_{=0} + \underbrace{\mathbb{E}[K_p(Y_1, Y_2)] - \mathbb{E}[K_p(X, Y)]}_{=0} = 0
\end{aligned}
$$

similarly, the same token can show that $h_{0,1}^p(Y_i) = 0$ and $h_{0,0}^p = 0$ under $H_0$ as well. Hence $L_{m,n}^p$ in (31) under $H_0$ can be simplified as:

$$
\begin{aligned}
L_{m,n}^p &= 3h_{0,0}^p - \frac{4}{m}\sum_{i=1}^m h_{1,0}^p(X_i) - \frac{4}{n}\sum_{j=1}^n h_{0,1}^p(Y_j) + \frac{4}{mn}\sum_{i=1}^m\sum_{j=1}^n h_{1,1}^p(X_i, Y_j) \\
&\quad + \binom{m}{2}^{-1}\sum_{0 \le i_1 < i_2 \le m} h_{2,0}^p(X_{i_1}, X_{i_2}) + \binom{n}{2}^{-1}\sum_{0 \le j_1 < j_2 \le m} h_{0,2}^p(Y_{j_1}, Y_{j_2}) \\
&\overset{H_0}{=} \binom{m}{2}^{-1}\sum_{0 \le i_1 < i_2 \le m} h_{2,0}^p(X_{i_1}, X_{i_2}) + \binom{n}{2}^{-1}\sum_{0 \le j_1 < j_2 \le n} h_{0,2}^p(Y_{j_1}, Y_{j_2}) + \frac{4}{mn}\sum_{i=1}^m\sum_{j=1}^n h_{1,1}^p(X_i, Y_j).
\end{aligned}
\tag{34}
$$

Through the above computation (34), the empirical estimator $\mathcal{E}_{m,n}^p = L_{m,n}^p$, under the null hypothesis, is completely expressed as linear combinations of $h_{2,0}^p$, $h_{0,2}^p$ and $h_{1,1}^p$. A beautiful strategy from (Gao & Shao, 2023) is to express these terms by a certain class of (infinite) basis functions, as if there three functions are "kernels" from the same source, which leads to simple computations of variance component $\mathrm{Var}(\mathcal{E}_{m,n}^p) = \mathrm{Var}(L_{m,n}^p)$.

Our goal is to recover their strategy this strategy of Euclidean spaces in the context of spheres, and to try finding adaptable bases in spherical harmonics of finite dimensional RKHS $H_p = \bigoplus_{i=0}^{2p}\mathcal{H}_i$ for large enough $p$. Recall that from (29), we have the following decompositions:

$$
\begin{cases}
h_{2,0}^p(X_1, X_2) &= K_p(X_1, X_2) + \mathbb{E}[K_p(Y_1, Y_2)] - \mathbb{E}_Y[K_p(X_1, Y)] - \mathbb{E}_Y[K_p(X_2, Y)]; \\[2mm]
h_{0,2}^p(Y_1, Y_2) &= \mathbb{E}[K_p(X_1, X_2)] + K_p(Y_1, Y_2) - \mathbb{E}_X[K_p(X, Y_1)] - \mathbb{E}_X[K_p(X, Y_2)]; \\[2mm]
h_{1,1}^p(X_1, Y_1) &= \mathbb{E}_X[K_p(X_1, X)] + \mathbb{E}_Y[K_p(Y_1, Y)] \\
&\quad -\frac{1}{2}K_p(X_1, Y_1) - \frac{1}{2}\mathbb{E}_Y[k_p(X_1, Y)] - \frac{1}{2}\mathbb{E}_X[K_p(X, Y_1)] - \frac{1}{2}\mathbb{E}[K_p(X, Y)].
\end{cases}
$$

Under the null hypothesis $H_0$ : $X \overset{d}{=} Y \overset{d}{=} Z$, take $Z_1$, $Z_2$ to be two independent copies of $Z$, then $h_{0,2}^p$, $h_{2,0}^p$ and $h_{1,1}^p$ can be rewritten under $H_0$ as follows:

$$
\begin{cases}
h_{2,0}^p(Z_1, Z_2) &= K_p(Z_1, Z_2) + \mathbb{E}[K_p(Y_1, Y_2)] - \mathbb{E}_Y[K_p(Z_1, Y)] - \mathbb{E}_Y[K_p(Z_2, Y)] \\
&\overset{H_0}{=} K_p(Z_1, Z_2) + \mathbb{E}[K_p(Z_1, Z_2)] - \mathbb{E}_Z[K_p(Z_1, Z)] - \mathbb{E}_Z[K_p(Z_2, Z)]; \\[2mm]
h_{0,2}^p(Z_1, Z_2) &= \mathbb{E}[K_p(X_1, X_2)] + K_p(Z_1, Z_2) - \mathbb{E}_X[K_p(X, Z_1)] - \mathbb{E}_X[K_p(X, Z_2)] \\
&\overset{H_0}{=} \mathbb{E}[K_p(Z_1, Z_2)] + K_p(Z_1, Z_2) - \mathbb{E}_Z[K_p(Z, Z_1)] - \mathbb{E}_Z[K_p(Z, Z_2)]; \\[2mm]
h_{1,1}^p(Z_1, Z_2) &= \mathbb{E}_X[K_p(Z_1, X)] + \mathbb{E}_Y[K_p(Z_2, Y)] \\
&\quad -\frac{1}{2}K_p(Z_1, Z_2) - \frac{1}{2}\mathbb{E}_Y[k_p(Z_1, Y)] - \frac{1}{2}\mathbb{E}_X[K_p(X, Z_1)] - \frac{1}{2}\mathbb{E}[K_p(X, Y)] \\
&\overset{H_0}{=} \mathbb{E}_Z[K_p(Z_1, Z)] + \mathbb{E}_Z[K_p(Z_2, Z)] \\
&\quad -\frac{1}{2}K_p(Z_1, Z_2) - \frac{1}{2}\mathbb{E}_Z[k_p(Z_1, Z)] - \frac{1}{2}\mathbb{E}_Z[K_p(Z, Z_2)] - \frac{1}{2}\mathbb{E}[K_p(Z_1, Z_2)] \\
&= -\frac{1}{2}K_p(Z_1, Z_2) + \frac{1}{2}\mathbb{E}_Z[K_p(Z_1, Z)] + \frac{1}{2}\mathbb{E}_Z[K_p(Z_2, Z)] - \frac{1}{2}\mathbb{E}[K_p(Z_1, Z_2)].
\end{cases}
\tag{35}
$$

From (35) above, one can instantly observe the following important functional equations among $h_{0,2}^p$, $h_{2,0}^p$ and $h_{1,1}^p$:

$$
h_{2,0}^p(Z_1, Z_2) \overset{H_0}{=} h_{0,2}^p(Z_1, Z_2) \overset{H_0}{=} -2h_{1,1}^p(Z_1, Z_2).
\tag{36}
$$

Also note that under the null hypothesis we have $\mathbb{E}[h_{2,0}^p(Z_1, Z_2)] = \mathbb{E}[h_{0,2}^p(Z_1, Z_2)] = 0$, and if we fix a particular value $z_1$, then it's easy to check that $\mathbb{E}[h_{2,0}^p(z_1, Z_2)] = \mathbb{E}[h_{0,2}^p(z_1, Z_2)] = 0$ for any $z_1 \in \mathbb{S}^d$. Therefore, it is natural to guess that $h_{2,0}^p(Z_1, Z_2)$ should be a "positive kernel function", *not* out of the whole $H_p = \bigoplus_{i=0}^{2p} \mathcal{H}_i$ (due to the vanishing expectation), *but* (probably) out of a subspace of $H_p$, but which subspace of $H_p = \bigoplus_{i=0}^{2p} \mathcal{H}_i$ should be the right choice?

**Definition A.4.** Let $Z$ be a random variable on $\mathbb{S}^d$ from the distribution under the null $H_0: X \overset{d}{=} Y \overset{d}{=} Z$, then let's define $H_Z^p := \{h \in H_p : \mathbb{E}[h(Z)] = 0\}$ as a sub-Hilbert space of $H_p = \bigoplus_{i=0}^{2p} \mathcal{H}_i$.

It is natural from classical intuition that $H_Z^p$ is a codimension one subspace of $H_p = \bigoplus_{i=0}^{2p} \mathcal{H}_i$. However, the probability distribution function of $Z$ might not be (in fact, very likely not at all) inside $H_p$ at all, hence a priori, one can even assume that $H_Z^p$ is even non-empty, but we can still give the following positive confirmation:

**Proposition A.5.** *If the probability distribution function of $Z$ comes from $L^2(\mathbb{S}^d)$, then $H_Z^p$ is indeed a non-empty codimension one subspace of $H_p = \bigoplus_{i=0}^{2p} \mathcal{H}_i$, namely $\dim(H_p) - \dim(H_Z^p) = 1$ for $p > 0$.*

*Proof.* Let $f(x)$ be the probability distribution function $Z$, and $f(x) \in L^2(\mathbb{S}^d)$. Consider the natural orthogonal projection map:

$$\Pi_p: L^2(\mathbb{S}^d) \subset \bigoplus_{i=0}^{\infty} \mathcal{H}_i \longrightarrow H_p = \bigoplus_{i=0}^{2p} \mathcal{H}_i, \tag{37}$$

Let $f_p$ be the image of this projection map, i.e. $\Pi_p(f) = f_p \in H_p = \bigoplus_{i=0}^{2p} \mathcal{H}_i$, then I claim that for any function $h \in H_p$, we have $< h, f_p >_{H_p} = < h, f >_{L^2(\mathbb{S}^d)}$. To prove this claim, notice that $\Pi_p(h) = h$ for any function $h \in H_p$, and note that projection operators are self-adjoint, then :

$$
\begin{aligned}
< h, f >_{L^2(\mathbb{S}^d)} &= < \Pi_p(h), f >_{L^2(\mathbb{S}^d)} \\
&= < h, \Pi_p(f) >_{L^2(\mathbb{S}^d)} \\
&= < h, f_p >_{H_p},
\end{aligned}
$$

thus the claim is concluded. Therefore the space $H_Z^p$ can be re-interpreted as:

$$H_Z^p = \{h \in H_p : \mathbb{E}[h(Z)] = 0\} = \{h \in H_p : < h, f_p >_{H_p} = \int_{\mathbb{S}^d} h f_p \mathrm{d}\omega = 0\}.$$

Note that $H_p$ is a finite dimensional Hilbert space, and $< \cdot, f_p >_{H_p}$, then it is an easy linear algebra fact that the vanishing set of a linear functional is a codimension one linear subspace of the entire finite dimensional Hilbert space $H_p$. $\square$

With Proposition A.5, we can give this sub-Hilbert space $H_Z^p$ a new inner product structure $< \cdot, \cdot >_Z$ by using the probability measure of $Z$ ($\mathrm{d}P := f\mathrm{d}\omega$ for $f$ a probability distribution function):

$$< h_1, h_2 >_Z := \int_{\mathbb{S}^d} h_1 h_2 \mathrm{d}P = \int_{\mathbb{S}^d} (h_1 h_2) f \mathrm{d}\omega = \int_{\mathbb{S}^d} (h_1 h_2) f_p \mathrm{d}\omega, \quad \forall h_1, h_2 \in H_Z^p \tag{38}$$

We hope to find an adaptable bases to span all three functions in (36), hence we need the Mercer's theorem to promise their existence, but in order to leverage this theorem, we need some positivity result:

**Proposition A.6.** *Let $d_p := \dim(H_p) = \dim(\bigoplus_{i=0}^{2p} \mathcal{H}_i)$ (the expression of $d_p$ can be found in Remark A.7), then under $H_0: X \overset{d}{=} Y \overset{d}{=} Z$, there exists an orthonormal bases $\{\phi_i^p\}_{i=1}^{d_p-1}$ with respect to the inner product $< \cdot, \cdot >_Z$ in (38), along with a sequence of nonnegative numbers $\{\lambda_i\}_{i=1}^{d_p-1} \subset \mathbb{R}$, such that*

$$h_{2,0}^p(Z_1, Z_2) = h_{0,2}^p(Z_1, Z_2) = \sum_{i=1}^{d_p-1} \lambda_i \phi_i(Z_1)\phi_i(Z_2).$$

*Proof.* This proposition essentially follows from the famous Mercer's theorem. However, in order to apply Mercer's theorem, we need to check the positivity condition, namely, we need to check the following inequality:

$$< \int_{\mathbb{S}^d} h_{2,0}^p(z_1,z_2)h(z_1)\mathrm{d}z_1, h(z_2) >_Z \geq 0, \quad \forall h \in H_Z^p;$$

by plugging $h_{2,0}^p(z_1,z_2)$ from (35) into the above inner product, we get the following summation:

$$
\begin{aligned}
&< \int_{\mathbb{S}^d} h_{2,0}^p(z_1,z_2)h(z_1)\mathrm{d}z_1, h(z_2) >_Z \\
=& \int_{\mathbb{S}^d}\int_{\mathbb{S}^d} h_{2,0}^p(z_1,z_2)h(z_1)h(z_2)f(z_2)\mathrm{d}z_1\mathrm{d}z_2 \\
=& \int_{\mathbb{S}^d}\int_{\mathbb{S}^d} [K_p(z_1,z_2)+\mathbb{E}[K_p(Z_1,Z_2)]-\mathbb{E}_Z[K_p(z_1,Z)]-\mathbb{E}_Z[K_p(z_2,Z)]]h(z_1)h(z_2)f(z_2)\mathrm{d}z_1\mathrm{d}z_2 \\
=& \int_{\mathbb{S}^d}\int_{\mathbb{S}^d} K_p(z_1,z_2)h(z_1)h(z_2)f(z_2)\mathrm{d}z_1\mathrm{d}z_2 + \mathbb{E}[K_p(Z_1,Z_2)]\int_{\mathbb{S}^d}\int_{\mathbb{S}^d} h(z_1)h(z_2)f(z_2)\mathrm{d}z_1\mathrm{d}z_2 \\
& - \int_{\mathbb{S}^d}\int_{\mathbb{S}^d} \mathbb{E}_Z[K_p(z_1,Z)]h(z_1)h(z_2)f(z_2)\mathrm{d}z_1\mathrm{d}z_2 - \int_{\mathbb{S}^d}\int_{\mathbb{S}^d} \mathbb{E}_Z[K_p(z_2,Z)]h(z_1)h(z_2)f(z_2)\mathrm{d}z_1\mathrm{d}z_2 \\
=& \int_{\mathbb{S}^d} h^2(z_2)f(z_2)\mathrm{d}z_2 + \mathbb{E}[K_p(Z_1,Z_2)]\cdot 0 - 2\int_{\mathbb{S}^d}\int_{\mathbb{S}^d} \mathbb{E}_Z[K_p(z_1,Z)]h(z_1)h(z_2)f(z_2)\mathrm{d}z_1\mathrm{d}z_2 \\
=& \mathbb{E}[h^2(Z_2)] - 2\int_{\mathbb{S}^d}\int_{\mathbb{S}^d}\int_{\mathbb{S}^d} K_p(z_1,z)h(z_1)h(z_2)f(z_2)f(z)\mathrm{d}z_1\mathrm{d}z_2\mathrm{d}z \\
=& \mathbb{E}[h^2(Z_2)] - 2\int_{\mathbb{S}^d}\int_{\mathbb{S}^d} h(z)h(z_2)f(z)f(z_2)\mathrm{d}z\mathrm{d}z_2 \\
=& \mathbb{E}[h^2(Z_2)] - 2\underbrace{\mathbb{E}[h(Z)]}_{=0}\underbrace{\mathbb{E}[h(Z_2)]}_{=0} \\
=& \mathbb{E}[h^2(Z_2)] \geq 0,
\end{aligned}
$$

hence the positivity follows. By general Mercer's theorem, both an orthonormal bases $\{\phi_i^p\}_{i=1}^{d_p-1}$ and the sequence $\{\lambda_i\}_{i=1}^{d_p-1}$ (with each $\lambda_i \geq 0$) exist such that $h_{2,0}^p(Z_1,Z_2) = h_{0,2}^p(Z_1,Z_2) = \sum_{i=1}^{d_p-1} \lambda_i\phi_i(Z_1)\phi_i(Z_2)$.

$\square$

*Remark* A.7. By spherical harmonics theory, our reproducing kernel $K_p$ has the interesting property that $K_p(z,z) = \left[\binom{d+2p}{d} + \binom{d+2p-1}{d}\right]/\mathrm{vol}(\mathbb{S}^d)$ for any $z \in \mathbb{S}^d$, a constant function on the sphere. Here the constant in the bracket $\binom{d+2p}{d} + \binom{d+2p-1}{d}$ is the same as $d_p$. Let us denote this constant number by $a_0^p$, then by (Gao & Shao, 2023) we can turn the estimator $\mathcal{V}_{m,n}^p$ in (19) into an unbiased estimator $\tilde{\mathcal{V}}_{m,n}^p$ for $V_p^2(Z)$ (16) by adding a bias correction term as below:

$$
\begin{aligned}
\tilde{\mathcal{V}}_{m,n}^p &:= \mathcal{V}_{m,n}^p - \frac{(a_0^p)^2}{(N-1)(N-3)} \\
&= \frac{1}{N(N-3)}\sum_{s\neq t}(_pA_{st}^{m,n})^2 - \frac{(a_0^p)^2}{(N-1)(N-3)};
\end{aligned}
$$

this bias correction term could be potentially helpful with small sample size, but when $N$ becomes much larger, the bias correction term will not be important.

The bases $\{\phi_i^p\}_{i=1}^{d_p-1}$ of $H_Z^p$ will be referred to as the "adaptable bases", which will be important for the calculation of the variance component $\mathrm{Var}(\mathcal{E}_{m,n}^p)$. Combining Proposition A.3.2 with the functional equation in (36), we can summarize the key result of this subsection in the following corollary:

**Corollary A.8.** *Under the null $H_0 : X \overset{d}{=} Y \overset{d}{=} Z$, there exists an adaptable bases $\{\phi_i\}_{i=1}^{d_p-1}$ in the Hilbert space $H_Z^p$ with respect to the inner product $< \cdot, \cdot >_Z$ defined in (38), in other words $\mathbb{E}[\phi_i(Z)\phi_j(Z)] = 0$ for $i \neq j$ and $\mathbb{E}[\phi_i^2(Z)] = 1$ with $\mathbb{E}[\phi_i(Z)] = 0$ for $1 \leq i \leq d_p - 1$, such that that the followings hold:*

☎ *Adaptable bases* $\{\phi_i\}_{i=1}^{d_p-1}$ *simultaneously expand all of* $h_{0,2}^p$, $h_{2,0}^p$ *and* $h_{1,1}^p$, *namely:*

$$
\begin{cases}
h_{2,0}^p(X_1, X_2) & = & \displaystyle\sum_{i=1}^{d_p-1} \lambda_i \phi_i(X_1)\phi_i(X_2) \\[2ex]
h_{0,2}^p(Y_1, Y_2) & = & \displaystyle\sum_{i=1}^{d_p-1} \lambda_i \phi_i(Y_1)\phi_i(Y_2) \\[2ex]
h_{1,1}^p(X, Y) & = & -\dfrac{1}{2}\displaystyle\sum_{i=1}^{d_p-1} \lambda_i \phi_i(X)\phi_i(Y);
\end{cases}
\tag{39}
$$

☎ *The non-negative sequence* $\{\lambda_i\}_{i=1}^{d_p-1}$ *satisfies the following equalities, in which* $\{Z_i\}_{i=1}^{3}$ *are independent identical copies of* $Z$:

$$
\begin{cases}
\displaystyle\sum_{i=1}^{d_p-1} \lambda_i & = & \mathbb{E}[h_{2,0}^p(Z, Z)] & = & \mathbb{E}[K_p(Z, Z)] - \mathbb{E}[K_p(Z_1, Z_2)] \\[3ex]
\displaystyle\sum_{i=1}^{d_p-1} \lambda_i^2 & = & \mathbb{E}[\{h_{2,0}^p(Z_1, Z_2)\}^2] & = & V_p^2(Z).
\end{cases}
\tag{40}
$$

*Proof.* The simultaneous expansion property easily follows from Proposition A.3.2 and the functional equation in (36).

As for the 2nd part, it is easy to se that $\sum_{i=1}^{d_p-1} \lambda_i = h_{2,0}^p(Z, Z)$ by using the orthonormality of the bases $\{\phi_i\}_{i=1}^{d_p-1}$ that expands $h_{2,0}^p$ in (39). On the other hand we can compute $h_{2,0}^p = (Z, Z)$ by plugging in $Z_1 = Z_2$ to (35), so we get:

$$
\begin{aligned}
h_{2,0}^p(Z_1, Z_1) & = & K_p(Z_1, Z_1) + \mathbb{E}[K_p(Z_1, Z_2)] - \mathbb{E}_Z[K_p(Z_1, Z)] - \mathbb{E}_Z[K_p(Z_1, Z)] \\
& = & K_p(Z_1, Z_1) - \mathbb{E}[K_p(Z_1, Z_2)];
\end{aligned}
$$

by taking the expectation on both sides of the above equality, we get

$$
\mathbb{E}[h_{2,0}^p(Z_1, Z_1)] = \mathbb{E}[K_p(Z_1, Z_1)] - \mathbb{E}[K_p(Z_1, Z_2)],
$$

thus the statement about $\sum_{i=1}^{d_p-1} \lambda_i$ follows.

To show the statement about $\sum_{i=1}^{d_p-1} \lambda_i^2$, it also comes from two different ways of computing the expectation $\mathbb{E}([h_{2,0}^p(Z_1, Z_2)]^2)$. The first way is to use the expansion (39) by the adaptable bases $\{\phi_i\}_{i=1}^{d_p-1}$:

$$
\begin{aligned}
\mathbb{E}[\{h_{2,0}^p(Z_1, Z_2)\}^2] & = & \mathbb{E}[(\sum_{i=1}^{d_p-1} \lambda_i \phi_i(Z_1)\phi_i(Z_2))^2] \\
& = & \sum_{1 \leq i,j \leq d_p-1} \lambda_i \lambda_j \mathbb{E}[(\phi_i(Z_1)\phi_i(Z_2))(\phi_j(Z_1)\phi_j(Z_2))] \\
& = & \sum_{1 \leq i,j \leq d_p-1} \lambda_i \lambda_j \mathbb{E}[\phi_i(Z_1)\phi_j(Z_1)]\mathbb{E}[\phi_i(Z_2)\phi_j(Z_2)] \\
& = & \sum_{1 \leq i=j \leq d_p-1} \lambda_i \lambda_j \mathbb{E}[\phi_i^2(Z_1)]\mathbb{E}[\phi_i^2(Z_2)] \\
& = & \sum_{1 \leq i \leq (d_p-1)} \lambda_i^2.
\end{aligned}
$$

On the other hand, by using the definition of $h_{2,0}^p$ from (29), we get the following expansion:

$$
\begin{aligned}
& [h_{2,0}^p(Z_1, Z_2)]^2 \\
= & \; k_p^2(Z_1, Z_2) + (\mathbb{E}[k_p(Z_2, Z_2)])^2 + (\mathbb{E}_Z[k_p(Z_1, Z)])^2 + (\mathbb{E}_Z[k_p(Z_2, Z)])^2 \\
& + 2k_p(Z_1, Z_2)\mathbb{E}[k_p(Z_1, Z_2)] - 2k_p(Z_1, Z_2)\mathbb{E}_Z[k_p(Z_1, Z)] - 2k_p(Z_1, Z_2)\mathbb{E}_Z[k_p(Z_2, Z)] \\
& - 2\mathbb{E}_Z[k_p(Z_1, Z_2)]\mathbb{E}_Z[k_p(Z_1, Z)] - 2\mathbb{E}_Z[k_p(Z_1, Z_2)]\mathbb{E}_Z[k_p(Z_2, Z)] \\
& + 2\mathbb{E}_Z[k_p(Z_1, Z)]\mathbb{E}_Z[k_p(Z_2, Z)].
\end{aligned}
\tag{41}
$$

By taking the expectation on both sides of (41), the right hand side only has three possibilities which are classified as below:

$$
\begin{cases}
(\mathbb{E}_Z[k_p(Z_1, Z)])^2 & \text{for} \quad \begin{aligned}[t] & (\mathbb{E}[k_p(Z_2, Z_2)])^2, \ k_p(Z_1, Z_2)\mathbb{E}[k_p(Z_1, Z_2)], \\ & \mathbb{E}_Z[k_p(Z_1, Z_2)]\mathbb{E}_Z[k_p(Z_1, Z)], \ \mathbb{E}_Z[k_p(Z_1, Z_2)]\mathbb{E}_Z[k_p(Z_2, Z)], \\ & \mathbb{E}_Z[k_p(Z_1, Z)]\mathbb{E}_Z[k_p(Z_2, Z)]; \end{aligned} \\[4mm]
\mathbb{E}[k_p^2(Z_1, Z_2)] & \text{for} \quad k_p^2(Z_1, Z_2); \\[4mm]
\mathbb{E}_Z[k_p(Z_1, Z)k_p(Z_2, Z)] & \text{for} \quad \begin{aligned}[t] & (\mathbb{E}_Z[k_p(Z_1, Z)])^2, \ E_Z[k_p(Z_2, Z)])^2, \\ & k_p(Z_1, Z_2)\mathbb{E}_Z[k_p(Z_1, Z)], \ k_p(Z_1, Z_2)\mathbb{E}_Z[k_p(Z_2, Z)]; \end{aligned}
\end{cases}
\tag{42}
$$

then by counting the coefficient of each three possibilities according to (42) after taking the expectation of (41), we get:

$$
\begin{aligned}
\mathbb{E}([h_{2,0}^p(Z_1, Z_2)]^2) &= \mathbb{E}[k_p^2(Z_1, Z_2)] + (1 + 1 - 2 - 2)\mathbb{E}_Z[k_p(Z_1, Z)k_p(Z_2, Z)] + (1 + 2 - 2 - 2 + 2)(\mathbb{E}_Z[k_p(Z_1, Z)])^2 \\
&= \mathbb{E}[k_p^2(Z_1, Z_2)] - 2\mathbb{E}_Z[k_p(Z_1, Z)k_p(Z_2, Z)] + (\mathbb{E}_Z[k_p(Z_1, Z)])^2 \\
&= V_p^2(Z);
\end{aligned}
$$

therefore from the two ways of computing $E([h_{2,0}^p(Z_1, Z_2)]^2)$, the statement about $\sum_{i=1}^{d_p-1} \lambda_i^2$ also follows.

$\square$

### A.3.3. CONCLUDING PROPOSITION 5.2

With the existence of adaptable bases under $H_0$ and reinterpret the $p$-th KSD variance as the sum of square $V_p^2(Z) = \sum_{i=1}^{d_p-1} \lambda_i^2$, we are finally ready to conclude the proof of Proposition 5.2.

At this point, the rest of the proof essentially uses the same idea from Section C.2 of (Gao & Shao, 2023), whose key idea is to utilize the simultaneous expansion property from Corollary A.8, although in their scenario where the Hilbert space was infinite dimensional while our Hilbert space $H_Z^p$ has finite dimension $d_p - 1$. To streamline our writing, we will include a short exposition of this proof.

THe key idea is to expand the "linear part" of $\mathcal{E}_{m,n}^p$ (which is $L_{m,n}^p$) with respect to the adaptable bases in Corollary A.8 under $H_0$; indeed $L_{m,n}^p$ is a linear combination of $\{h_{i,j}^p\}_{i+j\leq 2}$ in (34), while each term in $\{h_{i,j}^p\}_{i+j=2}$ was expanded by adaptable bases in Corollary A.8, hence a simple calculation shows that

$$
\begin{aligned}
L_{m,n}^p &\overset{H_0}{=} \binom{m}{2}^{-1} \sum_{0 \leq i_1 < i_2 \leq m} h_{2,0}^p(X_{i_1}, X_{i_2}) + \binom{n}{2}^{-1} \sum_{0 \leq j_1 < j_2 \leq n} h_{0,2}^p(Y_{j_1}, Y_{j_2}) + \frac{4}{mn} \sum_{i=1}^{m} \sum_{j=1}^{n} h_{1,1}^p(X_i, Y_j) \\
&= \binom{m}{2}^{-1} \sum_{0 \leq i_1 < i_2 \leq m} \sum_{k=1}^{d_p-1} \lambda_k \phi_k(X_{i_1})\phi_k(X_{i_2}) + \binom{n}{2}^{-1} \sum_{0 \leq j_1 < j_2 \leq n} \sum_{k=1}^{d_p-1} \lambda_k \phi_k(Y_{j_1})\phi_k(Y_{j_2}) \\
&\quad + \frac{4}{mn} \sum_{i=1}^{m} \sum_{j=1}^{n} (-\frac{1}{2}) \sum_{k=1}^{d_p-1} \phi_k(X_i)\phi_k(Y_j).
\end{aligned}
$$

Recall that under $H_0$ we combine these two samples into a bigger sample $\{Z_\ell\}_{\ell=1}^N$ with $N = m + n$:

$$
Z_\ell = \begin{cases} X_\ell, & 1 \leq \ell \leq m; \\ Y_{\ell-m}, & m < \ell \leq N. \end{cases}
$$

Notice that $i_1 \neq i_2$ and $j_1 \neq j_2$, then $L_{m,n}^p$ can be further rewritten in terms of $\{Z_\ell\}_{\ell=1}^N$:

$$
L_{m,n}^p \overset{H_0}{=} \sum_{\ell=1}^{N} \sum_{i=1}^{\ell-1} \sum_{k=1}^{d_p-1} \tau_{i,\ell,k}\phi(Z_i)\phi(Z_\ell), \text{ where } \tau_{i,\ell,k} \text{ depends on } i, \ell, \text{ and } \lambda_k;
$$

with this uniform expression of $L_{m,n}^p$ in terms of the combined sample $\{Z_\ell\}_{\ell=1}^N$, let $\psi_\ell = \sum_{i=1}^{\ell-1} \sum_{k=1}^{d_p-1} \tau_{i,\ell,k}\phi(Z_i)\phi(Z_\ell)$

and $\mathcal{F}_\ell = \sigma(Z_1, \cdots, Z_\ell)$, then it is easy to show that $\{\phi_\ell, \mathcal{F}_\ell\}_{\ell=1}^N$ forms a Martingale difference sequence, indeed:

$$
\begin{aligned}
\mathbb{E}[\psi_{\ell+1}|\mathcal{F}_\ell] &= \mathbb{E}[\sum_{i=1}^{\ell}\sum_{k=1}^{d_p-1}\tau_{i,\ell+1,k}\phi(Z_i)\phi(Z_{\ell+1})|\mathcal{F}_\ell] \\
&= \sum_{i=1}^{\ell}\sum_{k=1}^{d_p-1}\tau_{i,\ell+1,k}\phi(Z_i)\mathbb{E}[\phi(Z_{\ell+1})|Z_1,\cdots,Z_\ell] = \sum_{i=1}^{\ell}\sum_{k=1}^{d_p-1}\tau_{i,\ell+1,k}\phi(Z_i)\mathbb{E}[\phi(Z_{\ell+1})] = 0,
\end{aligned}
$$

then it is instant from this Martingale difference sequence $\{\phi_\ell, \mathcal{F}_\ell\}_{\ell=1}^N$, which gives the benefit of computing the variance component $\mathrm{Var}(L_{m,n}^p)$ in terms of the sum $\sum_{\ell=1}^N \mathbb{E}(\psi_\ell^2)$, but we need to compute each individual term $\mathbb{E}(\psi_\ell^2)$ first by using the orthonormality of the adaptable bases $\{\phi_k\}_{k=1}^{d_p-1}$:

$$
\begin{aligned}
\mathbb{E}(\psi_\ell^2) &= \sum_{1\le k_1,k_2\le d_p-1}\sum_{1\le i_1,i_2\le \ell-1}\tau_{i_1,\ell,k_1}\tau_{i_2,\ell,k_2}\mathbb{E}[\phi_{k_1}(Z_{i_1})\phi_{k_1}(Z_\ell)\phi_{k_2}(Z_{i_2})\phi_{k_2}(Z_\ell)] \\
&= \sum_{1\le k_1,k_2\le d_p-1}\sum_{1\le i_1,i_2\le \ell-1}\tau_{i_1,\ell,k_1}\tau_{i_2,\ell,k_2}\mathbb{E}[\phi_{k_1}(Z_{i_1})\phi_{k_2}(Z_{i_2})]\mathbb{E}[\phi_{k_1}(Z_\ell)\phi_{k_2}(Z_\ell)] \\
&= \sum_{k_1=k_2=k}\sum_{i_1=i_2=i}\tau_{i_1,\ell,k_1}\tau_{i_2,\ell,k_2} = \sum_{k=1}^{d_p-1}\sum_{i=1}^{\ell-1}\tau_{i,\ell,k}^2.
\end{aligned}
$$

Now by plugging into the Martingale difference sequence for the variance computation, we get:

$$
\begin{aligned}
\mathrm{Var}(L_{m,n}^p) &= \sum_{\ell=1}^N \mathbb{E}(\psi_\ell^2) = \sum_{k=1}^{d_p-1}\sum_{\ell=1}^N\sum_{i=1}^{\ell-1}\tau_{i,\ell,k}^2 \\
&= \sum_{k=1}^{d_p-1}(\sum_{1\le i<\ell\le N}\tau_{i,\ell,k}^2)
\end{aligned}
\tag{43}
$$

For a fixed $k$, there are three possibilities for $\tau_{i,\ell,k}$:

$$
\begin{cases}
Z_i \ \& \ Z_\ell \in \{X_i\}_{i=1}^m & \Rightarrow \quad \tau_{i,\ell,k} = \binom{m}{2}^{-1}\lambda_k \\[2mm]
Z_i \in \{X_i\}_{i=1}^m, Z_\ell \in \{Y_j\}_{j=1}^n \in & \Rightarrow \quad \tau_{i,\ell,k} = \dfrac{-2\lambda_k}{mn} \\[2mm]
Z_i \ \& \ Z_\ell \in \{Y_j\}_{j=1}^n & \Rightarrow \quad \tau_{i,\ell,k} = \binom{n}{2}^{-1}\lambda_k
\end{cases}
$$

Plugging in back (43), we get the following:

$$
\begin{aligned}
\mathrm{Var}(L_{m,n}^p) &= \sum_{k=1}^{d_p-1}(\sum_{1\le i<\ell\le N}\tau_{i,\ell,k}^2) \\
&= \sum_{k=1}^{d_p-1}[(\binom{m}{2}^{-1}\lambda_k)^2 \times \binom{m}{2} + (\frac{-2\lambda_k}{mn})^2 \times (mn) + (\binom{n}{2}^{-1}\lambda_k)^2 \times \binom{n}{2}] \\
&= c_{m,n}\sum_{k=1}^{d_p-1}\lambda_k^2 = c_{m,n}V_p^2(Z).
\end{aligned}
$$

### A.4. Proof of Theorem 5.5

For our previous proofs, we were proving results that were analogous to (Gao & Shao, 2023), however we were not able to directly plug in their result, but instead we had to write a parallel argument in the context of spherical harmonics.

Theorem 5.5 was analogous to Theorem 16 in (Gao & Shao, 2023), whose proof was in fact based upon very general stochastic analysis, without any particular assumptions of underlying space, so we can actually plug in their result to achieve our theorem. However, their theorem was deduced based upon several assumptions, so in order to utilize their work, we need to *validate* their assumptions in our context of spherical harmonics theory.

To state these assumptions, let's introduction the following notations: the key players are the different centered versions of the kernel $K_p(X_1, X_2)$; the fully centered version of this kernel is denoted by $d_p(X_1, X_2)$:

$$\begin{aligned} d_p(X_1, X_2) := K_p(X_1, X_2) - \mathbb{E}_{X_1}[K_p(X_1, X_2)] \\ - \mathbb{E}_{X_2}[K_p(X_1, X_2)] + \mathbb{E}[K_p(X_1, X_2)]; \end{aligned} \tag{44}$$

then based on this fully centered kernel $d_p(X_1, X_2)$, we can define the following functional:

$$\begin{aligned} & g_p(Z_1, Z_2, Z_3, Z_4) \\ =& d_p(Z_1, Z_2)d_p(Z_1, Z_3)d_p(Z_2, Z_4)d_p(Z_3, Z_4) \end{aligned} \tag{45}$$

Another centered version is the following partially centered kernel, denoted by $\bar{K}_p$:

$$\bar{K}_p(X_1, X_2) = K_p(X_1, X_2) - \mathbb{E}[K_p(X_1, X_2)]. \tag{46}$$

The following asymptotic condition based upon (46) is needed for both Lemma 5.3 and Theorem 5.5, for some $0 < \tau \leq 1$:

$$\frac{\mathbb{E}[|\bar{K}_p(Z_1, Z_2)|^{2+2\tau}]}{N^\tau [V_p^2(Z)]^{1+\tau}} \longrightarrow 0, \quad \text{when } N, p \to \infty. \tag{47}$$

Another asymptotic condition on the full centered kernel $d_p(Z_1, Z_2)$ is needed for the Theorem 5.5:

$$\frac{\{\mathbb{E}[g_p(Z_1, Z_2, Z_3, Z_4)]\}^{(1+\tau)/2}}{[V_p^2(Z)]^{1+\tau}} \longrightarrow 0, \quad \text{when } p \to \infty, \tag{48}$$

for some $0 < \tau \leq 1$.

In (Li et al., 2023), the authors defined the compositional kernel $k_p^\Gamma$, so one might hope to directly use this compositional kernel to perform non-parametric compositional two sample test, but we're not convinced that compositional kernels would satisfy the asymptotic growth condition in (48), hence the test statistic based on compositional kernels will not yield the studentization result as in Theorem 5.5.

We need to satisfy two asymptotic conditions of (47) and (48), which will be our workhorse of proving this theorem. Both conditions (47) and (48) requires that the condition holds for some $\tau \in (0, 1]$, but for simplicity, there's no harm to assume for the rest of this section that $\tau = 1$.

### A.4.1. CERTIFYING THE ASYMPTOTIC CONDITION (47) IN SPHERICAL HARMONICS

All terms in (47) are essentially expectations of kernel functions on several identically distributed and independent random variables. However, the partially centered kernel $\bar{K}_p(\cdot, \cdot)$ (46) has an extra term $\mathbb{E}[K_p(X_1, X_2)]$, and we want to show that this term has no asymptotic effect at all via the following:

$$\lim_{p \to \infty} \frac{\mathbb{E}(|\bar{K}_p(Z_1, Z_2)|^{2+2\tau})}{\mathbb{E}(|K_p(X_1, X_2)|^{2+2\tau})} = 1, \quad \text{for some } \tau \in (0, 1]. \tag{49}$$

As was remarked earlier, we can just take $\tau = 1$ and show that the quotient $\mathbb{E}(|\bar{K}_p(Z_1, Z_2)|^4)/\mathbb{E}(|K_p(X_1, X_2)|^4) \to 1$ as $p \to \infty$, but first of all we want to show that $\mathbb{E}[K_p(Z_1, Z_2)]$ is uniformly bounded above by a finite number for any positive $p$.

By the assumption of Theorem 5.5, the probability density function $P \in L^2(\mathbb{S}^d)$, then under the orthogonal projection map $\Pi_p$, denote the "projected measure" of $P$ under $\Pi_p$ as $P_p$, namely $P_p = \Pi_p(P)$. By the self-adjointness of projection

operator, we have the following:

$$
\begin{aligned}
\mathbb{E}[K_p(X_1, X_2)] &= \int_{\mathbb{S}^d} \int_{\mathbb{S}^d} K_p(x_1, x_2) P(x_1) P(x_2) \mathrm{d}x_1 \mathrm{d}x_2 \\
&= \int_{\mathbb{S}^d} P(x_2) \mathrm{d}x_2 \int_{\mathbb{S}^d} K_p(x_1, x_2) P(x_1) \mathrm{d}x_1 \\
&= \int_{\mathbb{S}^d} P(x_2) \mathrm{d}x_2 < K_p(\cdot, x_2), P(\cdot) >_{L^2(\mathbb{S}^d)} \\
&= \int_{\mathbb{S}^d} P(x_2) \mathrm{d}x_2 < \Pi_p(K_p(\cdot, x_2)), P(\cdot) >_{L^2(\mathbb{S}^d)} \\
&= \int_{\mathbb{S}^d} P(x_2) \mathrm{d}x_2 < K_p(\cdot, x_2), \Pi_p(P(\cdot)) >_{L^2(\mathbb{S}^d)} \\
&= \int_{\mathbb{S}^d} P(x_2) \mathrm{d}x_2 < K_p(\cdot, x_2), P_p(\cdot) >_{H_p} \\
&= \int_{\mathbb{S}^d} P(x_2) P_p(x_2) \mathrm{d}x_2;
\end{aligned}
\tag{50}
$$

if we let $p \to \infty$, then $P_p$ in $L^2$-norm on the sphere, then by (50]), we get:

$$
\mathbb{E}[K_p(X_1, X_2)] = \int_{\mathbb{S}^d} P(x_2) P_p(x_2) \mathrm{d}x_2 \overset{p \to \infty}{\longrightarrow} \int_{\mathbb{S}^d} P^2 \mathrm{d}x_2 = \|P\|^2_{L^2(\mathbb{S}^d)} < \infty.
$$

Uniform finiteness of $\mathbb{E}[K_p(X_1, X_2)]$ indicates that $\mathbb{E}[K_p(X_1, X_2)]$ should not impact much on the asymptotic behavior, but we still need to decompose $\mathbb{E}[|\bar{K}_p(Z_1, Z_2)|^4]$ can be decomposed in terms of $\mathbb{E}([K_p(Z_1, Z_2)]^m)$ ($m \leq 4$) and powers of $\mathbb{E}[K_p(X_1, X_2)]$, in order to carefully analyze other terms' growth with $p$:

$$
\begin{aligned}
\mathbb{E}[|\bar{K}_p(Z_1, Z_2)|^4] &= \mathbb{E}[\{K_p(Z_1, Z_2) - \mathbb{E}[K_p(X_1, X_2)]\}^4] \\
&= \mathbb{E}([K_p(Z_1, Z_2)]^4) - 4\mathbb{E}([K_p(Z_1, Z_2)]^3)\mathbb{E}[K_p(X_1, X_2)] \\
&\quad + 6\mathbb{E}([K_p(Z_1, Z_2)]^2)\mathbb{E}[K_p(X_1, X_2)]^2 - 4\mathbb{E}[K_p(Z_1, Z_2)]\mathbb{E}[K_p(X_1, X_2)]^3 \\
&\quad + \mathbb{E}[K_p(X_1, X_2)]^4
\end{aligned}
\tag{51}
$$

In order to estimate the asymptotic growth behavior of each term in (51), let's compute $\mathbb{E}[(K_p(Z_1, Z_2))^m]$ ($m \geq 2$):

$$
\begin{aligned}
\mathbb{E}([K_p(Z_1, Z_2)]^m) &= \int_{\mathbb{S}^d} \int_{\mathbb{S}^d} K_p^m(x_1, x_2) P(x_1) P(x_2) \mathrm{d}x_1 \mathrm{d}x_2 \\
&= \int_{\mathbb{S}^d} P(x_2) \mathrm{d}x_2 \int_{\mathbb{S}^d} K_p(x_1, x_2) P(x_1) K_p^{m-1}(x_1, x_2) \mathrm{d}x_1 \\
&= \int_{\mathbb{S}^d} P(x_2) \mathrm{d}x_2 < K_p(\cdot, x_2), P(\cdot) K_p^{m-1}(\cdot, x_2) > \\
&= \int_{\mathbb{S}^d} \Pi_p[P(x_2) K_p^{m-1}(x_2, x_2)] P(x_2) \mathrm{d}x_2 \\
&= \int_{\mathbb{S}^d} \Pi_p[P(x_2) d_p^{m-1} / \mathrm{vol}(\mathbb{S}^d)^{m-1}] P(x_2) \mathrm{d}x_2 \\
&= [\frac{d_p}{\mathrm{vol}(\mathbb{S}^d)}]^{m-1} \int_{\mathbb{S}^d} P(x_2) P_p(x_2) \mathrm{d}x_2,
\end{aligned}
\tag{52}
$$

where $d_p$ was defined in Proposition A.6, which is the dimension of $H_p = \bigoplus_{i=0}^{2p} \mathcal{H}_i$, and explicitly we can write down this dimension $d_p = \binom{d + 2p}{d} + \binom{d + 2p - 1}{d}$, therefore $\mathbb{E}([K_p(Z_1, Z_2)]^m) \sim d_p^{m-1}$ while $d_p \sim p^d$, hence in (51), the leading term $\mathbb{E}([K_p(Z_1, Z_2)]^4)$ has the fastest growth rate relative $p$, hence concluded (49); if we plug in $\tau = 1$, we have achieved the following:

$$
\lim_{p \to \infty} \frac{\mathbb{E}(|\bar{K}_p(Z_1, Z_2)|^4)}{\mathbb{E}(|K_p(X_1, X_2)|^4)} = 1.
\tag{53}
$$

The denominator of (47) is involved with the term $V_p^2(Z)$, whose definition is recalled as follows:

$$
V_p^2(Z) = \mathbb{E}[K_p^2(Z_1, Z_2)] - 2\mathbb{E}[K_p(Z_1, Z_2) K_p(Z_1, Z_3)] + \left(\mathbb{E}[K_p(Z_1, Z_2)]\right)^2;
$$

in the above expression, we've already shown $\mathbb{E}[K_p(Z_1, Z_2)]$ is uniformly finite, and here we claim that the middle term $\mathbb{E}[K_p(Z_1, Z_2)K_p(Z_1, Z_3)]$ is also uniformly finite, independent of $p$:

$$
\begin{aligned}
\mathbb{E}[K_p(Z_1, Z_2)K_p(Z_1, Z_3)] &= \int_{\mathbb{S}^d}\int_{\mathbb{S}^d}\int_{\mathbb{S}^d} K_p(z_1, z_2)K_p(z_1, z_3)P(z_1)P(z_2)P(z_3)\mathrm{d}z_1\mathrm{d}z_2\mathrm{d}z_3 \\
&= \int_{\mathbb{S}^d} P(z_3)\mathrm{d}z_3 \int_{\mathbb{S}^d} P(z_2)\mathrm{d}z_2 \int_{\mathbb{S}^d} K_p(z_1, z_2)K_p(z_1, z_3)P(z_1)\mathrm{d}z_1 \\
&= \int_{\mathbb{S}^d} P(z_3)\mathrm{d}z_3 \int_{\mathbb{S}^d} P(z_2)\mathrm{d}z_2 < K_p(\cdot, z_2), K_p(\cdot, z_3)P(\cdot) > \\
&= \int_{\mathbb{S}^d} P(z_3)\mathrm{d}z_3 \int_{\mathbb{S}^d} P(z_2)\Pi_p[K_p(z_2, z_3)P(z_2)]\mathrm{d}z_2 \\
&= \int_{\mathbb{S}^d} P(z_3)\mathrm{d}z_3 \int_{\mathbb{S}^d} \Pi_p[P(z_2)]K_p(z_2, z_3)P(z_2)\mathrm{d}z_2 \ \ (\Pi_p \text{ is self-adjoint}) \\
&= \int_{\mathbb{S}^d} P(z_3)\mathrm{d}z_3 < K_p(\cdot, z_3), \Pi_p[P(\cdot)]P(\cdot) > \\
&= \int_{\mathbb{S}^d} P(z_3)\mathrm{d}z_3 < K_p(\cdot, z_3), \Pi_p[P(\cdot)P(\cdot)] > \ \ (\Pi_p\big[\Pi_p[P(\cdot)]P(\cdot)\big] = \Pi_p[P(\cdot)P(\cdot)]) \\
&= \int_{\mathbb{S}^d} \Pi_p[P^2(z_3)]P(z_3)\mathrm{d}z_3 \overset{p\to\infty}{\longrightarrow} \int_{\mathbb{S}^d} P^3 \mathrm{d}z_3;
\end{aligned}
$$
(54)

note that the condition of Theorem 5.5 assumed that $P$ is continuous on the sphere, a compact space, so naturally $P \in L^3(\mathbb{S}^d)$, hence by (54) the middle term $\mathbb{E}[K_p(Z_1, Z_2)K_p(Z_1, Z_3)]$ is uniformly finite independent $p$, hence we have arrived at the following:

$$
\lim_{p\to\infty} \frac{\mathbb{E}[K_p^2(Z_1, Z_2)]}{V_p^2(Z)} = 1.
$$
(55)

Now, combining (53) and (55), Condition (47) is equivalent to the following for $\tau = 1$:

$$
\frac{\mathbb{E}[K_p^4(Z_1, Z_2)]}{N(\mathbb{E}[K_p^2(Z_1, Z_2)])^2} \to 0, \ \text{when } N, p \to \infty \text{ and } \lim_{N\to\infty} \frac{p^d}{N} = 0.
$$
(56)

Now, let us compute the limit in (56): by (52), the numerator in (56) $\mathbb{E}[K_p^4(Z_1, Z_2)] \sim d_p^3$, the denominator of (56) contains the term $(\mathbb{E}[K_p^2(Z_1, Z_2)])^2 \sim d_p^2$, so note that $d_p \sim p^d$ the whole ratio of (56) is asymptotically the same as:

$$
\frac{\mathbb{E}[K_p^4(Z_1, Z_2)]}{N(\mathbb{E}[K_p^2(Z_1, Z_2)])^2} \sim \frac{d_p^3}{Nd_p^2} = \frac{d_p}{N} \sim \frac{p^d}{N};
$$

however, recall that our assumption assumes that $p^d/N \to 0$ as $N \to \infty$, therefore

$$
\frac{\mathbb{E}[K_p^4(Z_1, Z_2)]}{N(\mathbb{E}[K_p^2(Z_1, Z_2)])^2} \sim \frac{p^d}{N} \longrightarrow 0, \ \text{when } N \to \infty
$$

as desired.

### A.4.2. CERTIFYING CONDITION (48) IN SPHERICAL HARMONICS

Condition (48) involves the fully centered kernel $d_p(\cdot, \cdot)$ of $K_p(\cdot, \cdot)$ as in (44) which is recalled as below:

$$
d_p(Z_1, Z_2) = K_p(Z_1, Z_2) - \mathbb{E}_{Z_1}[K_p(Z_1, Z_2)] - \mathbb{E}_{Z_2}[K_p(Z_1, Z_2)] + \mathbb{E}[K_p(Z_1, Z_2)];
$$

out of the above fully centered kernels, another complicated functional $g_p(Z_1, Z_2, Z_3, Z_4)$ in (45) is also involved in Condition (48 recalled as below:

$$
g_p(Z_1, Z_2, Z_3, Z_4) = d_p(Z_1, Z_2)d_p(Z_1, Z_3)d_p(Z_2, Z_4)d_p(Z_3, Z_4).
$$

Due to the fact that there are too many centered terms involved, we need to simplify our computation by getting rid of centered terms asymptotically, which is a strategy that was already used Section (A.4.1). Luckily, this strategy still works due to the following lemma:

**Lemma A.9.** *Under the null hypothesis $X \overset{d}{=} Y \overset{d}{=} Z$ with the combined data set $\{Z_\ell\}_{\ell=1}^N$ along with assumptions in Theorem 5.5, the following asymptotic condition holds:*

$$\lim_{p \to \infty} \frac{\mathbb{E}[K_p(Z_1, Z_2) K_p(Z_1, Z_3) K_p(Z_2, Z_4) K_p(Z_3, Z_4)]}{\mathbb{E}[g_p(Z_1, Z_2, Z_3, Z_4)]} = 1.$$

*Proof.* One can expand $g_p(Z_1, Z_2, Z_3, Z_4)$ by plugging into the following

$$d_p(Z_i, Z_j) = K_p(Z_i, Z_j) - \mathbb{E}_{Z_i}[K_p(Z_i, Z_j)] - \mathbb{E}_{Z_j}[K_p(Z_i, Z_j)] + \mathbb{E}[K_p(Z_i, Z_j)],$$

whose those centered terms are $\mathbb{E}_{Z_i}[K_p(Z_i, Z_j)]$, $\mathbb{E}_{Z_j}[K_p(Z_i, Z_j)]$ and $\mathbb{E}[K_p(Z_i, Z_j)]$ along with the non-centered term $K_p(Z_i, Z_j)$. We can classify the terms by using the following notations for $1 \leq i < j \leq 4$:

$$\begin{cases} h_0^p(Z_i, Z_j) &= \quad K_p(Z_i, Z_j); \\[1mm] h_1^p(Z_i, Z_j) &= \quad \mathbb{E}_{Z_i}[K_p(Z_i, Z_j)]; \\[1mm] h_2^p(Z_i, Z_j) &= \quad \mathbb{E}_{Z_j}[K_p(Z_i, Z_j)]; \\[1mm] h_3^p(Z_i, Z_j) &= \quad \mathbb{E}[K_p(Z_i, Z_j)]; \end{cases}$$

Then with the above notation, whenever expanding $g_p(Z_1, Z_2, Z_3, Z_4)$ by plugging into $d_p(Z_i, Z_j)$, we will get a linear combination of the following form:

$$h_{p_1}^p(Z_1, Z_2) h_{p_2}^p(Z_1, Z_3) h_{p_3}^p(Z_2, Z_4) h_{p_4}^p(Z_3, Z_4). \tag{57}$$

The first key observation for this lemma comes from the following claim: If $\sum_{i=1}^4 p_i > 0$, then the expectation $\mathbb{E}[h_{p_1}^p(Z_1, Z_2) h_{p_2}^p(Z_1, Z_3) h_{p_3}^p(Z_2, Z_4) h_{p_4}^p(Z_3, Z_4)]$ is uniformly bounded above by a finite number which is independent of $p$. From Claim A.4.2, the only term that can go to infinity among the terms that show up in the expansion of $g_p(Z_1, Z_2, Z_3, Z_4)$ is when all $p_1 = p_2 = p_3 = p_4 = 0$, which is the term $\mathbb{E}[K_p(Z_1, Z_2) K_p(Z_1, Z_3) K_p(Z_2, Z_4) K_p(Z_3, Z_4)]$.

Now going back to Claim A.4.2: in order to show this claim, it essentially can be done by case-to-case analysis, which is pretty similar to the computations done in Section A.4.1. Here we will only show one case as a demonstration example where $p_1 = 1$ and $p_2 = p_3 = p_4 = 0$, namely to compute the following:

$$\mathbb{E}[h_1^p(Z_1, Z_2) h_0^p(Z_1, Z_3) h_0^p(Z_2, Z_4) h_0^p(Z_3, Z_4)] = \mathbb{E}[\mathbb{E}_{Z_1}[K_p(Z_1, Z_2)] K_p(Z_1, Z_3) K_p(Z_2, Z_4) K_p(Z_3, Z_4)].$$

Notice that the projection operator $\Pi_p[\Pi_p[P]Q] = \Pi_p[PQ]$ for $P, Q \in L^2(\mathbb{S}^d)$, and by repeated use of self-adjointness of

the projection operator $\Pi_p$, similar to the tricks used in (54), we can proceed our computation:

$$
\begin{aligned}
&\mathbb{E}[\mathbb{E}_{Z_1}[K_p(Z_1, Z_2)]K_p(Z_1, Z_3)K_p(Z_2, Z_4)K_p(Z_3, Z_4)]\\
&= \int_{\mathbb{S}^d}\int_{\mathbb{S}^d}\int_{\mathbb{S}^d}\int_{\mathbb{S}^d} \mathbb{E}_{Z_1}[K_p(Z_1, z_2)]K_p(z_1, z_3)K_p(z_2, z_4)K_p(z_3, z_4)P(z_1)P(z_2)P(z_3)P(z_4)\mathrm{d}z_1\mathrm{d}z_2\mathrm{d}z_3\mathrm{d}z_4\\
&= \int_{\mathbb{S}^d}\int_{\mathbb{S}^d}\int_{\mathbb{S}^d} K_p(z_1, z_3)K_p(z_3, z_4)P(z_1)P(z_3)P(z_4)\mathrm{d}z_1\mathrm{d}z_3\mathrm{d}z_4 \int_{\mathbb{S}^d} \mathbb{E}_{Z_1}[K_p(Z_1, z_2)]K_p(z_2, z_4)P(z_2)\mathrm{d}z_2\\
&= \int_{\mathbb{S}^d}\int_{\mathbb{S}^d}\int_{\mathbb{S}^d} K_p(z_1, z_3)K_p(z_3, z_4)P(z_1)P(z_3)P(z_4)\Pi_p\big[\mathbb{E}_{Z_1}[K_p(Z_1, z_4)]P(z_4)\big]\mathrm{d}z_1\mathrm{d}z_3\mathrm{d}z_4\\
&= \int_{\mathbb{S}^d}\int_{\mathbb{S}^d} K_p(z_1, z_3)P(z_1)P(z_3)\mathrm{d}z_1\mathrm{d}z_3 \int_{\mathbb{S}^d} K_p(z_3, z_4)\Pi_p\big[\mathbb{E}_{Z_1}[K_p(Z_1, z_4)]P(z_4)\big]P(z_4)\mathrm{d}z_4\\
&= \int_{\mathbb{S}^d}\int_{\mathbb{S}^d} K_p(z_1, z_3)P(z_1)P(z_3)\Pi_p\big[\mathbb{E}_{Z_1}[K_p(Z_1, z_3)]P^2(z_3)\big]\mathrm{d}z_1\mathrm{d}z_3\\
&= \int_{\mathbb{S}^d} P(z_1)\mathrm{d}z_1 \int_{\mathbb{S}^d} K_p(z_1, z_3)P(z_3)\Pi_p\big[\mathbb{E}_Z[K_p(Z, z_3)]P^2(z_3)\big]\mathrm{d}z_3 \quad (Z \overset{d}{=} Z_1)\\
&= \int_{\mathbb{S}^d} \Pi_p\big[\mathbb{E}_Z[K_p(Z, z_1)]P^3(z_1)\big]P(z_1)\mathrm{d}z_1\\
&= \int_{\mathbb{S}^d} \mathbb{E}_Z[K_p(Z, z_1)]P^3(z_1)\Pi_p[P(z_1)]\mathrm{d}z_1 \quad (\Pi_p \text{ is self-adjoint})\\
&= \int_{\mathbb{S}^d}\int_{\mathbb{S}^d} K_p(z, z_1)P(z)P^3(z_1)\Pi_p[P(z_1)]\mathrm{d}z_1\mathrm{d}z\\
&= \int_{\mathbb{S}^d} P(z)\mathrm{d}z \int_{\mathbb{S}^d} K_p(z, z_1)P^3(z_1)\Pi_p[P(z_1)]\mathrm{d}z_1\\
&= \int_{\mathbb{S}^d} P(z)\Pi_p[P^4(z)]\mathrm{d}z;
\end{aligned}
\tag{58}
$$

therefore, $\mathbb{E}[\mathbb{E}_{Z_1}[K_p(Z_1, Z_2)]K_p(Z_1, Z_3)K_p(Z_2, Z_4)K_p(Z_3, Z_4)] \longrightarrow \|P\|_{L^5(\mathbb{S}^d)}^5$, which is finite (and independent of $p$) according to the assumption of Theorem 5.5. All of the rest of the uniform finiteness cases for (57) with $\sum_{i=1}^4 p_i > 0$ can be checked by similar computations.

For a general summary why the expectation for $h_{p_1}^p(Z_1, Z_2)h_{p_2}^p(Z_1, Z_3)h_{p_3}^p(Z_2, Z_4)h_{p_4}^p(Z_3, Z_4)$ is uniformly finite when $\sum_{i=1}^4 p_i > 0$, see Remark (A.10).

With the uniform finiteness for $\sum_{i=1}^4 p_i > 0$, our lemma follows from the following claim:

$$
\mathbb{E}[h_0^p(Z_1, Z_2)h_0^p(Z_1, Z_3)h_0^p(Z_2, Z_4)h_0^p(Z_3, Z_4)] \sim d_p \ \text{ as } p \to \infty.
$$

The proof of Claim A.4.2 will be given Proposition (A.11), whose proof will be given in that proposition. Now we explain how this lemma follows from Claim A.4.2 and Claim A.4.2, which is very straightforward: indeed, the denominator can be expressed as the sum:

$$
\mathbb{E}[g_p(Z_1, Z_2, Z_3, Z_4)] = \sum_{0 \leq p_1, p_2, p_3, p_4 \leq 3} \mathbb{E}[h_{p_1}^p(Z_1, Z_2)h_{p_2}^p(Z_1, Z_3)h_{p_3}^p(Z_2, Z_4)h_{p_4}^p(Z_3, Z_4)];
\tag{59}
$$

in the above decomposition, every term $\mathbb{E}[h_{p_1}^p(Z_1, Z_2)h_{p_2}^p(Z_1, Z_3)h_{p_3}^p(Z_2, Z_4)h_{p_4}^p(Z_3, Z_4)]$ is uniformly finite independent of $p$ for all $\sum_{i=1}^4 p_i > 0$, and there are only finite many terms with $\sum_{i=1}^4 p_i > 0$.

When $p_i = 0$ for every $1 \leq i \leq 4$, then by Claim A.4.2 we have $\mathbb{E}[h_0^p(Z_1, Z_2)h_0^p(Z_1, Z_3)h_0^p(Z_2, Z_4)h_0^p(Z_3, Z_4)] \sim d_p$, the dominant term of the sum (59), then the denominator $\mathbb{E}[g_p(Z_1, Z_2, Z_3, Z_4)] \sim d_p$ as $p \to \infty$. On the other hand, the numerator is just $\mathbb{E}[h_0^p(Z_1, Z_2)h_0^p(Z_1, Z_3)h_0^p(Z_2, Z_4)h_0^p(Z_3, Z_4)] \sim d_p$, the same asymptotic growth as the denominator for $p \to \infty$, hence our lemma follows

$\square$

*Remark* A.10. One might wonder whether there is a uniform reason why such finite result holds for all $\sum_{i=1}^4 p_i > 0$ cases. The answer is yes, and we give a high-level heuristic overview why the finiteness holds if any $p_i > 0$ for $1 \leq i \leq 4$: therefore for kernels $K_p(z_1, z_2)]$, $K_p(z_1, z_3)$ $K_p(z_2, z_4)$ and $K_p(z_3, z_4)$ involved, and each integral $\int$ will "consume" a kernel, with all together four integrals for the expectation of each term in (57).

However, when there is any $p_i > 0$ for $1 \le i \le 4$, that means some (partial) expectation was already taken for some $K_p(z_i, z_j)$, in addition to the four integrals for the expectation of each term; note that taking (partial) expectation is equivalent to another "integral", which implies that there are at least five integrals involved; before taking the final integral, no kernel functions are left, while the remaining functions are only coming from powers of (truncated) probability density functions, which by our assumptions of the theorem, do not contribute to infinite values in the final integral.

**Proposition A.11.** *The following term goes to infinity together the degree parameter $p \to \infty$, in particular:*

$$\mathbb{E}[K_p(Z_1, Z_2)K_p(Z_1, Z_3)K_p(Z_2, Z_4)K_p(Z_3, Z_4)] \sim d_p = \binom{d+2p}{d} + \binom{d+2p-1}{d}, \; as \; p \to \infty.$$

*Proof.* By using the same technique in (58), we can compute this expectation as follows:

$$
\begin{aligned}
&\mathbb{E}[K_p(Z_1, Z_2)K_p(Z_1, Z_3)K_p(Z_2, Z_4)K_p(Z_3, Z_4)] \\
&= \int_{\mathbb{S}^d} \int_{\mathbb{S}^d} \int_{\mathbb{S}^d} \int_{\mathbb{S}^d} K_p(z_1, z_2)K_p(z_1, z_3)K_p(z_2, z_4)K_p(z_3, z_4)P(z_1)P(z_2)P(z_3)P(z_4)\mathrm{d}z_1\mathrm{d}z_2\mathrm{d}z_3\mathrm{d}z_4 \\
&= \int_{\mathbb{S}^d} \int_{\mathbb{S}^d} \int_{\mathbb{S}^d} K_p(z_1, z_3)K_p(z_3, z_4)P(z_1)P(z_3)P(z_4)\mathrm{d}z_1\mathrm{d}z_3\mathrm{d}z_4 \int_{\mathbb{S}^d} K_p(z_1, z_2)K_p(z_2, z_4)P(z_2)\mathrm{d}z_2 \\
&= \int_{\mathbb{S}^d} \int_{\mathbb{S}^d} \int_{\mathbb{S}^d} K_p(z_1, z_3)K_p(z_3, z_4)P(z_1)P(z_3)P(z_4)\Pi_p\big[K_p(z_1, z_4)P(z_4)\big]\mathrm{d}z_1\mathrm{d}z_3\mathrm{d}z_4 \\
&= \int_{\mathbb{S}^d} \int_{\mathbb{S}^d} K_p(z_1, z_3)P(z_1)P(z_3)\mathrm{d}z_1\mathrm{d}z_3 \int_{\mathbb{S}^d} K_p(z_3, z_4)\Pi_p\big[K_p(z_1, z_4)P(z_4)\big]P(z_4)\mathrm{d}z_4 \\
&= \int_{\mathbb{S}^d} \int_{\mathbb{S}^d} K_p(z_1, z_3)P(z_1)P(z_3)\Pi_p\big[K_p(z_1, z_3)P^2(z_3)\big]\mathrm{d}z_1\mathrm{d}z_3 \\
&= \int_{\mathbb{S}^d} P(z_1)\mathrm{d}z_1 \int_{\mathbb{S}^d} K_p(z_1, z_3)P(z_3)\Pi_p\big[K_p(z_1, z_3)P^2(z_3)\big]\mathrm{d}z_3 \\
&= \int_{\mathbb{S}^d} \Pi_p\big[K_p(z_1, z_1)P^3(z_1)\big]P(z_1)\mathrm{d}z_1 \\
&= d_p \int_{\mathbb{S}^d} \Pi[P^3(z_1)]P(z_1)\mathrm{d}z_1;
\end{aligned}
$$
(60)

the final integral term in (60) goes to $\|P\|_{L^4(S^4)}^4$ as $p \to \infty$, which is finite by our theorem's assumption, hence we have got:

$$\mathbb{E}[K_p(Z_1, Z_2)K_p(Z_1, Z_3)K_p(Z_2, Z_4)K_p(Z_3, Z_4)] \sim d_p, \; as \; p \to \infty;$$

therefore, this proposition follows.

$\square$

To conclude the Condition (48), note that Lemma A.9 simplifies the condition (for $\tau = 1$) as follows:

$$\frac{\mathbb{E}[K_p(Z_1, Z_2)K_p(Z_1, Z_3)K_p(Z_2, Z_4)K_p(Z_3, Z_4)]}{[V_p^2(Z)]^2} \longrightarrow 0, \; as \; p \to \infty.$$
(61)

It suffices to verify the simplified Condition (61), which is very easy to do based on what we have developed so far. Indeed, the numerator, by Proposition A.11, is asymptotically equivalent to $d_p$, while the denominator, according to (55) is asymptotically equivalent to $\left(\mathbb{E}[K_p^2(Z_1, Z_2)]\right)^2$ which is also asymptotically the same as $d_p^2$ by (52) for $p \to \infty$. Therefore the denominator is asymptotically more dominating ($d_p^2$) than the numerator ($d_p$), hence Condition (61) holds, therefore Condition (48) holds, thus concluding our proofs.

# B. Details of Experiments

## B.1. Overview

Our goal is to compare our proposed method with baseline models using the compositional and directional data. Each compositional data sample is produced by a Dirichlet–Multinomial (DM) data-generating process (DGP) that naturally creates zeros in the observed counts.

We include the following files:

- `dgp_dm_two_sample.py`: Dirichlet–Multinomial DGP.

- `empirical_size.py`: Empirical size under $H_0$.

- `empirical_power.py`: Empirical power under $H_1$.

- `compositional_simulation.py`: Master file for combining results from the above three files.

- `README.md`: Providing detailed instruction to reproduce the simulation results.

- `compositional-and-directional-two-sample-test.zip`: All the files are available in this file.

## B.2. DGP file: `dgp_dm_two_sample.py`

Generate two independent samples from a DM model and normalize to the simplex. The DGP produces zeros in *counts*; compositions may contain zeros before CLR preprocessing.

**Inputs.**

- Dimension: $d$.

- Sample sizes: $(m, n)$ for group 0 and group 1.

- Simplex means: $\mu_0, \mu_1 \in \Delta^{d-1}$ (strictly positive entries required by Dirichlet).

- Concentration base: $\kappa_{\text{base}} > 0$.

- Concentration multiplier (group 1): $\eta > 0$.

- Library size: either (i) fixed $N$, or (ii) calibrated $N$ to target a zero-fraction $p_z$ (if you use the calibration version).

- RNG seed: `seed`.

For group $g \in \{0, 1\}$ define Dirichlet concentration vectors

$$\alpha_0 = \kappa_{\text{base}}\mu_0, \qquad \alpha_1 = \kappa_{\text{base}}\eta\mu_1.$$

For each sample:

$$\pi \sim \text{Dirichlet}(\alpha_g), \qquad c \mid \pi \sim \text{Multinomial}(N, \pi), \qquad x = c/\sum_{j=1}^{d} c_j.$$

The output compositions $x$ lie on the simplex; zeros can appear in $c$ and thus in $x$.

### B.3. Empirical size for compositional data: `empirical_size.py`

Estimate empirical size under

$$H_0: \ P_X = P_Y \quad \text{(implemented as } \mu_0 = \mu_1, \ \eta = 1 \text{ in the DM model).}$$

Size is calculated as rejection rates at $\alpha \in \{0.01, 0.05, 0.10\}$. We reported only the result for $\alpha = 0.05$.

We compute four pooled-matrix statistics:

- **CLR-ED**: ED on CLR-transformed data.

- **CLR-MMD**: Gaussian MMD on CLR space (bandwidth via median heuristic).

- **SPH-$\sqrt{\cdot}$-p2**, **SPH-$\sqrt{\cdot}$-p4**: Studentized spherical energy distance $T_{p,mn}$ on $\sqrt{x}$ with $p \in \{2, 4\}$.

- **SPH-L2-p2**, **SPH-L2-p4**: Studentized spherical energy distance $T_{p,mn}$ on $x/\|x\|_2$ with $p \in \{2, 4\}$.

Since CLR requires strictly positive entries, we apply a pseudocount:

$$x \leftarrow \frac{x + \varepsilon}{\sum_j (x_j + \varepsilon)}, \qquad \varepsilon = \texttt{eps\_clr},$$

then compute

$$\text{clr}(x) = \log x - \frac{1}{d} \sum_{j=1}^{d} \log x_j.$$

A permutation draw re-indexes the pooled matrix into two groups. The p-value is

$$p = \frac{1 + \#\{b : T_b \geq T_{\text{obs}}\}}{B + 1},$$

with $B = \texttt{B\_perm}$ permutations. We used $\texttt{B\_perm} = 200$ for our simulation.

### B.4. Empirical power for compositional data: `empirical_power.py`

The empirical power under $H_1 : P \neq Q$ is calculated using the following formula:

$$\text{Power} = Pr(T > c_\alpha),$$

where $T$ is a test statistic, including our statistics $T_{m,n}^p$, and $c_\alpha$ is the critical value at significance level at $\alpha$.

Estimate empirical power under alternatives that modify group 1 by: a mean shift ($\mu_1 \neq \mu_0$) and/or a concentration change ($\eta \neq 1$).

- $\mu$:rare - Log ratio mean shift concentrated on the two least prevalent components, increasing their relative abundance while decreasing the two most abundant

- $\mu$:abundant - Log ratio mean shift concentrated on the two most abundant components, increasing their relative abundance while decreasing the two rarest.

- $\mu$:uniform - Dense log ratio mean shift distributed across all components

- concentration - No mean shift and instead the Dirichlet concentration parameter is scaled by $\eta$, making sample $X$ more overdispersed ($\eta > 1$) relative to sample $Y$

Rejection rates at $\alpha \in \{0.01, 0.05, 0.10\}$ can be computed across $R = 200$ Monte Carlo replicates. We reported only the results for $\delta = 0.05$ and $\eta = 2$.

## B.5. Directional Statistics

The simulation study for the directional statistics is fully specified in Section 6.2. Figure 5 demonstrates the empirical distributional shapes and locations of our test statistics under the null and alternative hypotheses. The key observation is that as the sample size increases, the null distribution becomes approximately normal and the alternative distribution separates farther from the null distribution.

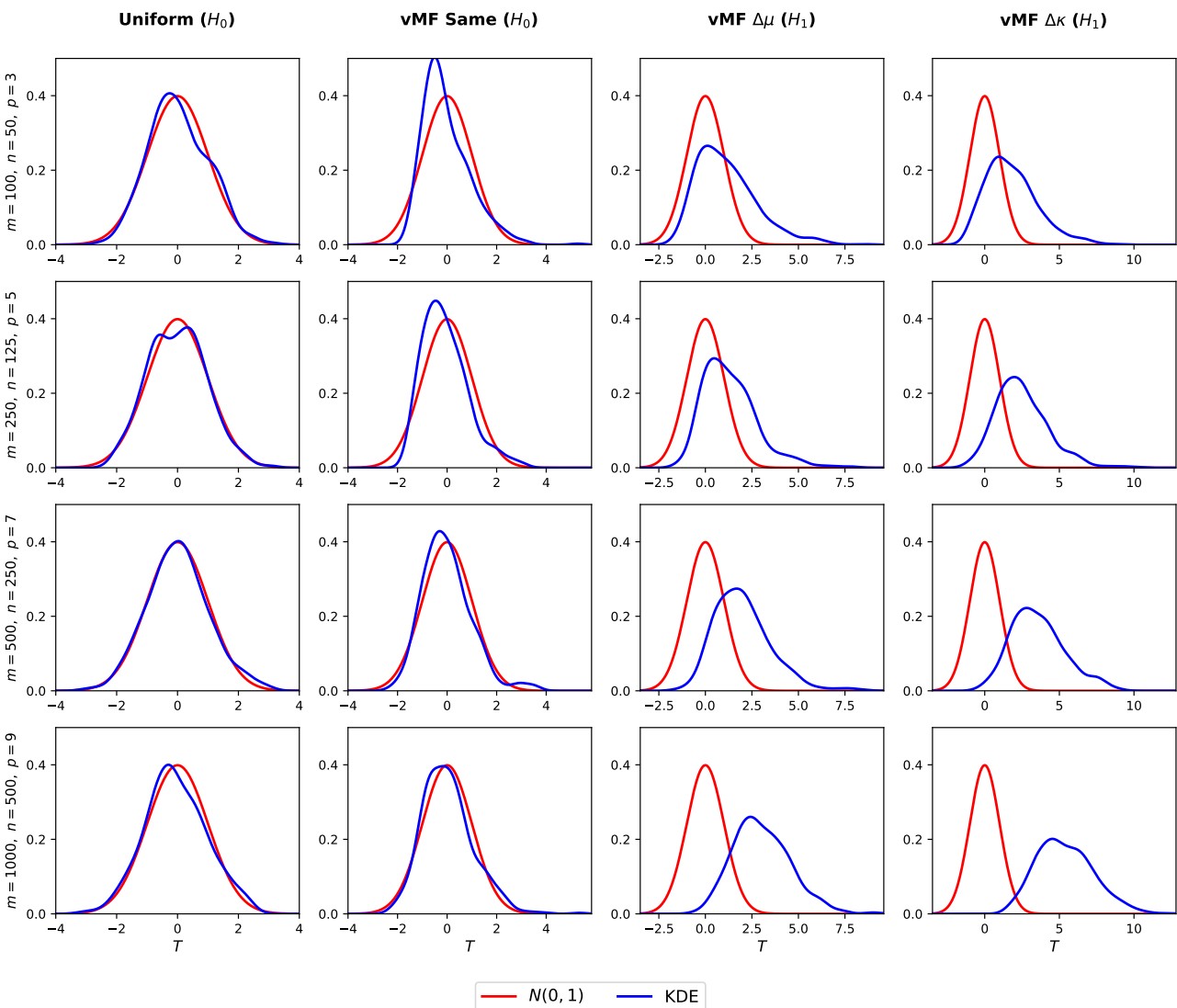

*Figure 5.* KDE of the studentized test statistic (blue) overlaid with the $N(0,1)$ distribution (red). The first two columns show convergence to $N(0,1)$ under $H_0$ as sample size increases. For the alternatives in columns 3 and 4, we observe increasing distance from the null distribution as the sample size increases from the top row ($m = 100, n = 50$) to the bottom row ($m = 1000, n = 500$). Traversing down a column represents increasing sample size.

**Experiment with von Mises Kernels**

The setup for this experiment is that both samples are generated by the vMF distribution, but the alternative is true by construction. Here the means are identical but the concentration parameters $\kappa$ are different and in the ratio of 1.75. Thus the differences between the distributions arises solely due to differences in the concentrations about the mean. Specifically, for the data generating proceeses $\kappa_X = 5 \max(d/2, 1)$ and $\kappa_Y = 8.75 \max(d/2, 1)$, so that $(\kappa_X, \kappa_Y) = (5, 8.75)$ at $d = 2$, $(12.5, 21.875)$ at $d = 5$. The vMF-MMD bandwidth is set by the median heuristic as $1/\operatorname{median}(d_{\text{geo}})$, where $d_{\text{geo}}$ denotes pairwise geodesic distances in the pooled sample, yielding an approximate value of $1.5$ across configurations. We compare our statistic SPH-$\sqrt{}$-p2 to the vMF kernel, which is a natural choice for directional data (Xu & Matsuda, 2020).

According to the following Table 2, it is clear that **our proposed test statistic has higher power** for all samples sizes and data dimensions we explored. Our method has competitive size control under the null hypothesis $H_0$, but is slightly inflated. This is because our test statistic is studentized, which magnifies the difference between the distributions when the distributions are very close. So observing a slightly inflated (but still competitive) Type I error rate along with higher power for our method is in line with our theoretical expectations. Note we report permutation p-values for our test statistic in addition to the asymptotic p-values (i.e. the inference framework based on our new CLT) out of curiosity and for interested readers.

*Table 2.* Empirical Type I error (H0) and power (H1) for vMF-MMD and SPH-$\sqrt{}$-p2 across dimensions and sample sizes.

| $d$ | $m = n$ | $H_0$ | | | $H_1$ | | |
|---|---|---|---|---|---|---|---|
| | | vMF-MMD | SPH-$\sqrt{}$-p2 perm | SPH-$\sqrt{}$-p2 asymp | vMF-MMD | SPH-$\sqrt{}$-p2 perm | SPH-$\sqrt{}$-p2 asymp |
| 2 | 25 | 0.050 | 0.050 | 0.067 | 0.133 | 0.200 | 0.257 |
| 2 | 50 | 0.063 | 0.050 | 0.063 | 0.273 | 0.390 | 0.477 |
| 2 | 100 | 0.047 | 0.057 | 0.073 | 0.607 | 0.697 | 0.753 |
| 5 | 25 | 0.067 | 0.060 | 0.073 | 0.167 | 0.393 | 0.473 |
| 5 | 50 | 0.053 | 0.047 | 0.073 | 0.407 | 0.740 | 0.790 |
| 5 | 100 | 0.073 | 0.070 | 0.087 | 0.860 | 0.980 | 0.987 |

Additionally, we performed a sensitivity analysis to understand the power of the vMF-MMD as a function of its bandwidth parameter. We note that even in the optimal regime where the bandwidth parameter is 5 or 10, our test statistic has higher power. The results are recorded in Figure 6, and the top performing vMF-MMD tests from that figure are recorded in Table 3 for curious readers.

Here are the full results of the sensitivity analysis. Note that our test statistic has higher power across the entirety of the range of bandwidth values.

*Table 3.* Empirical Type I error (H0) and power (H1) comparison between vMF-based methods (bandwidth 5, 10) and our method across dimensions and sample sizes.

| | | $H_0$ | | | $H_1$ | | |
|---|---|---|---|---|---|---|---|
| $d$ | $m = n$ | vMF-5 | vMF-10 | SPH-$\sqrt{\cdot}$-p2 | vMF-5 | vMF-10 | SPH-$\sqrt{\cdot}$-p2 |
| 2 | 25 | 0.040 | 0.033 | 0.040 | 0.183 | 0.170 | 0.233 |
| 2 | 50 | 0.060 | 0.053 | 0.070 | 0.370 | 0.367 | 0.477 |
| 2 | 100 | 0.050 | 0.040 | 0.077 | 0.687 | 0.653 | 0.743 |
| 5 | 25 | 0.063 | 0.063 | 0.080 | 0.360 | 0.427 | 0.473 |
| 5 | 50 | 0.047 | 0.057 | 0.077 | 0.683 | 0.720 | 0.773 |
| 5 | 100 | 0.063 | 0.057 | 0.097 | 0.963 | 0.963 | 0.987 |

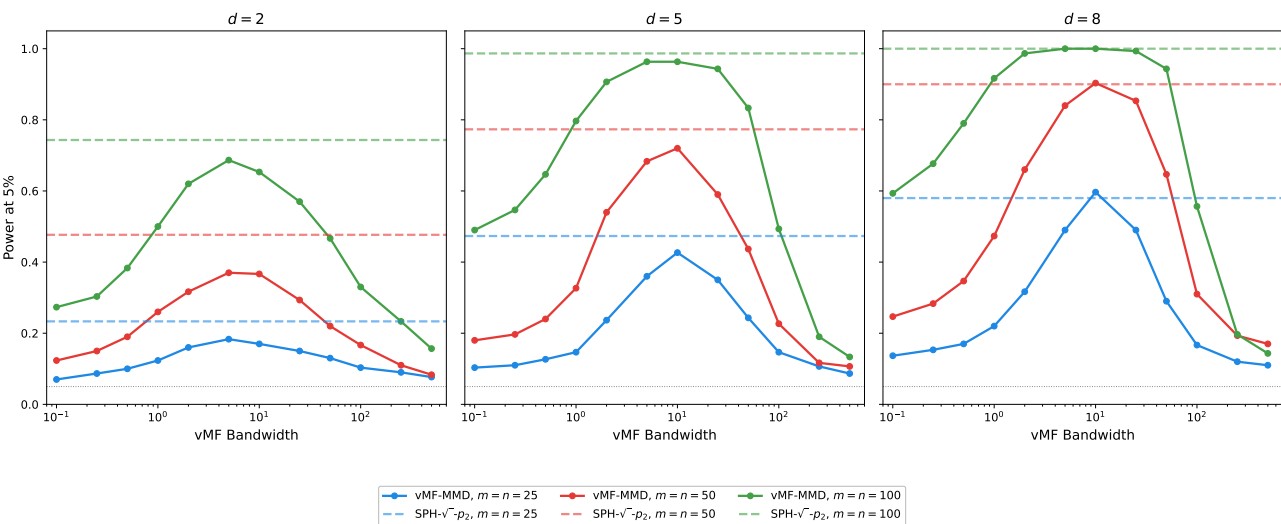

*Figure 6.* Sensitivity study of the vMF-MMD bandwidth parameter. Power at $\alpha = 0.05$ versus kernel bandwidth ($d = 2$ left, $d = 5$ center, $d = 8$ right; vMF concentration alternative, $\kappa$ ratio 1.75). Solid: vMF-MMD, $m = n \in \{25, 50, 100\}$ (blue/red/green). Dashed: proposed SPH-$\sqrt{\cdot}$-$p_2$ (bandwidth-free), same sample sizes. For example, to interpret these results, note that the green curves correspond to the indicated sample size regime, the green dashed line is the (bandwidth independent) performance of our method, the solid green curve is the performance of the vMF-MMD as a function of the bandwidth parameter, etc. Note our test statistic has higher power over the majority of the range of bandwidth values, and in many scenarios has higher power over the entire range. When $d = 8$, Condition 1 in Lemma 5.3 is violated, but our method is still competitive to the best bandwidth parameter even if our condition is not met.

## B.6. Additional Simulations and Real Data Analysis

### B.6.1. ADDITIONAL SIMULATION EXPERIMENTS

We performed extensive simulations with the DM distribution described in Appendix B.2. In all of these simulations $\delta = 0.3$, $\eta = 2.0$, and $\alpha = .05$. We observe that when the data dimension $d$ is small (equal to 10 here) and the differences between the two samples arises due to the abundant taxa (the $\mu$: rare alternative) then the version of our test statistic using the L2 normalization is the best choice (SPH-L2-p2 here). This is a little intuitive in that the L2 normalization preserves the ratio between the prevalence of the dominant taxa, whereas the square root transformation will magnify the noise in the smallest components (rarest taxa). When the signal is in the rarest taxa (the $\mu$ rare alternative), the square root transformation magnifies this difference more than other taxa. Notice that overall, the version of our test statistic with the $\sqrt{\cdot}$ transformation and setting $p \sim log(d)$ is the most robust choice, as it has the highest power in the majority of scenarios. The results are presented in Table 4. As can be seen, under the null hypothesis, all methods show empirical rejection rates close to the nominal 5% level across all dimensions and sample sizes. For the rare-taxon mean alternative, all methods suffer from low empirical power, but in other scenarios, our proposed method is highly competitive. In particular, our proposed test outperforms other methods in empirical power for the concentration alternative. Noticeably, our test shows substantially high power at $d = 100$ with $m = n = 25$ for the uniform mean-shift and concentration alternatives, even though the

compositional dimension substantially exceeds the sample sizes, which may imply that our method could be effective in some HDLSS situations.

*Table 4.* Empirical rejection rates at $\alpha = 0.05$ across varying dimension $d$, sample sizes $(m, n)$, and sparsity levels (pz).

| $d$ | $m$ | $n$ | pz | Test Statistic | Null | $\mu$: rare | $\mu$: abundant | $\mu$: uniform | concentration |
|---|---|---|---|---|---|---|---|---|---|
| 10 | 25 | 25 | 6% | CLR-ED | 0.05 | 0.16 | 0.04 | 0.22 | 0.14 |
| 10 | 25 | 25 | 6% | CLR-MMD | 0.06 | 0.15 | 0.04 | 0.21 | 0.14 |
| 10 | 25 | 25 | 6% | HEL-MMD | 0.06 | 0.14 | 0.62 | 1.00 | 0.16 |
| 10 | 25 | 25 | 6% | SPH-√-p2 | 0.07 | 0.18 | 0.69 | 1.00 | 0.13 |
| 10 | 25 | 25 | 6% | SPH-√-p4 | 0.07 | 0.17 | 0.66 | 1.00 | 0.23 |
| 10 | 25 | 25 | 6% | SPH-L2-p2 | 0.05 | 0.07 | 0.80 | 1.00 | 0.17 |
| 10 | 25 | 25 | 6% | SPH-L2-p4 | 0.04 | 0.07 | 0.70 | 0.99 | 0.27 |
| 10 | 50 | 50 | 6% | CLR-ED | 0.05 | 0.33 | 0.07 | 0.47 | 0.18 |
| 10 | 50 | 50 | 6% | CLR-MMD | 0.06 | 0.34 | 0.05 | 0.44 | 0.16 |
| 10 | 50 | 50 | 6% | HEL-MMD | 0.06 | 0.29 | 0.99 | 1.00 | 0.28 |
| 10 | 50 | 50 | 6% | SPH-√-p2 | 0.09 | 0.36 | 0.99 | 1.00 | 0.16 |
| 10 | 50 | 50 | 6% | SPH-√-p4 | 0.09 | 0.33 | 0.99 | 1.00 | 0.42 |
| 10 | 50 | 50 | 6% | SPH-L2-p2 | 0.08 | 0.12 | 1.00 | 1.00 | 0.27 |
| 10 | 50 | 50 | 6% | SPH-L2-p4 | 0.08 | 0.12 | 0.99 | 1.00 | 0.56 |
| 30 | 25 | 25 | 11% | CLR-ED | 0.04 | 0.07 | 0.07 | 0.50 | 0.45 |
| 30 | 25 | 25 | 11% | CLR-MMD | 0.04 | 0.08 | 0.07 | 0.48 | 0.43 |
| 30 | 25 | 25 | 11% | HEL-MMD | 0.03 | 0.04 | 0.26 | 1.00 | 0.63 |
| 30 | 25 | 25 | 11% | SPH-√-p2 | 0.05 | 0.04 | 0.28 | 1.00 | 0.57 |
| 30 | 25 | 25 | 11% | SPH-√-p4 | 0.05 | 0.03 | 0.27 | 1.00 | 0.96 |
| 30 | 25 | 25 | 11% | SPH-L2-p2 | 0.05 | 0.05 | 0.37 | 1.00 | 0.75 |
| 30 | 25 | 25 | 11% | SPH-L2-p4 | 0.07 | 0.06 | 0.32 | 1.00 | 0.98 |
| 30 | 50 | 50 | 11% | CLR-ED | 0.06 | 0.11 | 0.05 | 0.88 | 0.82 |
| 30 | 50 | 50 | 11% | CLR-MMD | 0.06 | 0.12 | 0.05 | 0.88 | 0.80 |
| 30 | 50 | 50 | 11% | HEL-MMD | 0.08 | 0.05 | 0.48 | 1.00 | 0.99 |
| 30 | 50 | 50 | 11% | SPH-√-p2 | 0.09 | 0.06 | 0.57 | 1.00 | 0.98 |
| 30 | 50 | 50 | 11% | SPH-√-p4 | 0.10 | 0.06 | 0.52 | 1.00 | 1.00 |
| 30 | 50 | 50 | 11% | SPH-L2-p2 | 0.06 | 0.06 | 0.70 | 1.00 | 1.00 |
| 30 | 50 | 50 | 11% | SPH-L2-p4 | 0.08 | 0.04 | 0.56 | 1.00 | 1.00 |
| 100 | 25 | 25 | 54% | CLR-ED | 0.03 | 0.05 | 0.08 | 0.91 | 0.09 |
| 100 | 25 | 25 | 54% | CLR-MMD | 0.03 | 0.05 | 0.08 | 0.91 | 0.09 |
| 100 | 25 | 25 | 54% | HEL-MMD | 0.03 | 0.04 | 0.10 | 0.97 | 0.36 |
| 100 | 25 | 25 | 54% | SPH-√-p2 | 0.05 | 0.04 | 0.10 | 0.98 | 0.91 |
| 100 | 25 | 25 | 54% | SPH-√-p4 | 0.04 | 0.05 | 0.09 | 0.89 | 1.00 |
| 100 | 25 | 25 | 54% | SPH-L2-p2 | 0.05 | 0.06 | 0.09 | 0.78 | 0.79 |
| 100 | 25 | 25 | 54% | SPH-L2-p4 | 0.06 | 0.08 | 0.07 | 0.38 | 0.79 |
| 100 | 50 | 50 | 54% | CLR-ED | 0.06 | 0.04 | 0.07 | 1.00 | 0.15 |
| 100 | 50 | 50 | 54% | CLR-MMD | 0.06 | 0.04 | 0.07 | 1.00 | 0.15 |
| 100 | 50 | 50 | 54% | HEL-MMD | 0.05 | 0.03 | 0.11 | 1.00 | 0.86 |
| 100 | 50 | 50 | 54% | SPH-√-p2 | 0.06 | 0.05 | 0.12 | 1.00 | 1.00 |
| 100 | 50 | 50 | 54% | SPH-√-p4 | 0.05 | 0.06 | 0.11 | 1.00 | 1.00 |
| 100 | 50 | 50 | 54% | SPH-L2-p2 | 0.06 | 0.06 | 0.12 | 0.99 | 1.00 |
| 100 | 50 | 50 | 54% | SPH-L2-p4 | 0.06 | 0.04 | 0.08 | 0.80 | 0.99 |

B.6.2. REAL DATA APPLICATION

.

We utilized the throat microbiome dataset (Charlson et al., 2010), which is widely used for two-sample compositional data testing. The data contain two groups of subjects (28 smokers and 32 nonsmokers) with 856 operational taxonomic units. We compared our method with the selected two-sample test methods and reported the results in Table 5. In addition to using all 856 operational taxonomic units, we explored prevalence filtering to reduce the dimensionality of the dataset. For the rows with **threshold** column equal to 20 percent we dropped OTU columns where the sparsity was less than this amount. Since this real microbiome dataset is very sparse, this has the effect of dropping the number of feature columns (OTUs) from 856 to 114. After removing the appropriate OTUs the data was renormalized to the simplex (i.e. the sum of each row is 1). After

prevalence filtering the sparsity naturally decreases. Out of theoretical interest we include permutation p-values for our new test statistic in addition to the asymptotic p-values. Notice that all methods reject the null hypothesis at significance level $\alpha = 0.05$. In fact the majority of methods reject the null hypothesis even at the significance level of $\alpha = 0.01$. There appears to be a real difference in the microbiomes of smokers versus nonsmokers, as suggested by these results in Table 5.

*Table 5.* Test statistics and corresponding asymptotic and permutation p-values across sparsity thresholds. Dashes indicate unavailable asymptotic p-values.

| threshold | $d$ | pz | Method | statistic | p-val (asymp) | p-val (perm) |
|---|---|---|---|---|---|---|
| 0% | 856 | 89.4% | CLR-ED | 2.5103 | – | 0.0015 |
| 0% | 856 | 89.4% | CLR-MMD | 0.0119 | – | 0.0015 |
| 0% | 856 | 89.4% | HEL-MMD | 0.0378 | – | 0.0005 |
| 0% | 856 | 89.4% | SPH-$\sqrt{}$-p2 | 5.2042 | 0.0000 | 0.0010 |
| 0% | 856 | 89.4% | SPH-$\sqrt{}$-p4 | 4.5606 | 0.0000 | 0.0005 |
| 0% | 856 | 89.4% | SPH-$\sqrt{}$-p6 | 3.9933 | 0.0000 | 0.0015 |
| 0% | 856 | 89.4% | SPH-L2-p2 | 3.7148 | 0.0001 | 0.0040 |
| 0% | 856 | 89.4% | SPH-L2-p4 | 3.3033 | 0.0005 | 0.0030 |
| 0% | 856 | 89.4% | SPH-L2-p6 | 3.0044 | 0.0013 | 0.0070 |
| 20% | 114 | 50.4% | CLR-ED | 2.5072 | – | 0.0055 |
| 20% | 114 | 50.4% | CLR-MMD | 0.0184 | – | 0.0055 |
| 20% | 114 | 50.4% | HEL-MMD | 0.0447 | – | 0.0010 |
| 20% | 114 | 50.4% | SPH-$\sqrt{}$-p2 | 4.9528 | 0.0000 | 0.0010 |
| 20% | 114 | 50.4% | SPH-$\sqrt{}$-p4 | 3.9658 | 0.0000 | 0.0015 |
| 20% | 114 | 50.4% | SPH-$\sqrt{}$-p6 | 3.1364 | 0.0009 | 0.0040 |
| 20% | 114 | 50.4% | SPH-L2-p2 | 3.7172 | 0.0001 | 0.0055 |
| 20% | 114 | 50.4% | SPH-L2-p4 | 3.2089 | 0.0007 | 0.0080 |
| 20% | 114 | 50.4% | SPH-L2-p6 | 2.7576 | 0.0029 | 0.0130 |

