# OpenReview forum: "A Studentized Spherical Harmonics–Based Nonparametric Two-Sample Test for Compositional and Directional Data"
_ICML.cc/2026/Conference — ICML 2026 regular_

### Official Review · Reviewer_cpPZ · 2026-03-11

**Soundness:** 3
**Presentation:** 3
**Significance:** 3
**Originality:** 3
**Overall Recommendation:** 4
**Confidence:** 5

**Summary:**

This paper extends the kernel-based two-sample testing framework of Gao and Shao (2023) to compositional and directional data in non-Euclidean spaces. Using the topological equivalence among several compositional domains established in Li et al.(2023), including the simplex, the positive orthant of the sphere, and the spherical quotient space, the compositional two-sample testing problem is reformulated as a distribution comparison problem on the sphere $\mathbb{S}^d$. The authors construct spherical kernels using spherical harmonics and define a kernel energy distance, which leads to a two-sample U-statistic. Following the studentization idea of Gao and Shao (2023), they construct a studentized test statistic via KSD covariance estimation and establish its asymptotic normality when both the sample size and the harmonic truncation order $p$ diverge under a non-Euclidean geometric framework.

**Compliance With Llm Reviewing Policy:**

Affirmed.

**Key Questions For Authors:**

1 Lines 232--235 mention that permutation or bootstrap procedures may suffer from potential inconsistency.
Since the proposed method is based on spherical kernels and the test statistic is formulated as a U-statistic,
I find the mechanism leading to bootstrap inconsistency in the present framework somewhat unclear, and it would be helpful if the authors could provide additional explanation.

2 In particular, the paper cites Zhu and Shao (2021), where permutation-based energy distance or MMD tests may exhibit trivial power in high-dimensional Euclidean settings when the data dimension diverges.
However, in the current paper the data dimension $d$ is fixed and the asymptotics are driven by the harmonic truncation order $p \to \infty$.
These appear to correspond to different regimes.
Therefore, I am curious whether the inconsistency phenomenon discussed in Zhu and Shao (2021) is also relevant in the present setting, or whether the diverging truncation order $p$ may lead to a similar issue.

3 The simulation study compares several statistics, including CLR-ED, CLR-MMD, and multiple SPH-based variants (e.g., SPH-$\sqrt{\cdot}$-p2, SPH-$\sqrt{\cdot}$-p4, SPH-L2-p2, SPH-L2-p4).
However, the rationale for including these particular methods and variants is not fully explained.
It would be helpful if the authors could clarify the purpose of comparing these statistics and discuss how practitioners should choose among these alternatives in practice.


 4 In Table~1, the empirical sizes of several proposed methods appear to be somewhat higher than the nominal level $0.05$. Could the authors provide additional simulation results under a wider range of sample sizes and compositional dimensions, for example: $N\in\{50,100,200,400\}$, with different sample proportions $m/N=\rho$, where $\rho\in(0,1)$. And $d+1\in\{5,10,15,20\}$.


5 The asymptotic theory requires the harmonic truncation order $p \to \infty$,
while in finite-sample settings $p$ must be fixed.
In the simulations, several SPH-based statistics corresponding to different truncation orders (e.g., $p=2,4$) are reported, but the rationale for choosing these values is not clearly discussed.
Could the authors clarify how the truncation order $p$ should be selected in practice and explain the motivation for the choices used in the experiments?


6 The theoretical analysis assumes continuous distributions on $\mathbb{S}^d$. However, compositional data in applications such as microbiome studies are often generated from Dirichlet–Multinomial models (counts followed by normalization), which produce discrete distributions and zero components. Would the authors consider evaluating the method under such data-generating mechanisms?

7 While the theoretical development is interesting, it would be helpful to include a real data example
(e.g., microbiome compositional data or other directional datasets) to illustrate the practical relevance
of the proposed method and to demonstrate how the method behaves in realistic applications.

**Limitations:**

yes

**Strengths And Weaknesses:**

Strengths

1 The paper leverages the topological equivalence results in Li et al. (2023) to map compositional data on the simplex to directional data on the unit sphere. This representation avoids the limitations of log-ratio transformations, which cannot handle compositional data containing zero-valued components.

2 Following the idea of Gao and Shao (2023), the authors construct a studentized U-statistic and derive the asymptotic null distribution of the test statistic. This enables the proposed test to use analytic critical values instead of relying on permutation or bootstrap procedures, providing a theoretically tractable inference framework.


Weaknesses

1 A large portion of the paper is devoted to geometric background, while the core statistical methodology appears relatively late. I would suggest streamlining the presentation so that the main statistical ideas are introduced earlier.

2 The simulation study is relatively limited. To better assess the finite-sample behavior of the proposed test, I suggest considering additional settings; see Questions 3--5 below for specific suggestions.

---

> ### Author Rebuttal · Authors · 2026-03-31
>
> In this response, the supplementary material can be found via the following anonymous url
> "https://anonymous-public-assets.s3.us-west-2.amazonaws.com/30744AppendixB.pdf", the updated codes are available upon request.
>
> Q1&2: Our method does not suffer from an inconsistency issue. In Zhu & Shao the inconsistency is ultimately due to the distributions being too similar (i.e., when the marginal distributions are identical), which is the difficulty. Our fix via studentization is to “magnify” the distance of two distributions by dividing the “variance of the distances"; indeed in Proposition 5.2 (page 7), the term of c_{m,n} in the variance goes to zero as sample size grows, which does magnify the kernel energy distances (KED).
>
> It does seem a little confusing that both MMD-permutation and our constructions, but our motivation of using U-statistics is to theoretically derive the variance of KED, which is used to studentize the KED, and surprisingly studentizations give asymptotically normal test statistics under the null hypothesis.
>
> Our method uses studentization to magnify the differences between distributions to handle this case and others that led to inconsistency there. Our test statistic requires the degree p to go to infinity in theory; in practice, we need to truncate “p” to a finite level. Here the real issue is to find a large enough “p” so that even the numerator starts detecting the difference of two distributions,  but practically, we do not need very large “p” to tell the two apart, as the norms of “p-th” term converges to 0 in Fourier expansion (spherical harmonics is high-dim generalization of Fourier expansions). As long as we find a small p to detect small differences, the denominator will magnify it, therefore the same issue is unlikely to occur for us.
>
>
>
> Q3: While we observe that our test statistic performs better with the L2 normalization to map from the simplex to the first
> orthant of the sphere when the dimension is small and the signal lies in the dominant taxa (μ abundant alternative), the most
> robust choice across varying dimension, sample size, and type of alternative is to select the version where we use the square
> root transformation. The heuristic of choosing p ∼ log(d) works well. Thus as a practical prescription for practioners we recommend to use SPH-√-pn where p ∼ log(d). In our experiments SPH-√-p4 was the most robust choice.
>
> Q4: We performed additional n and d (especially small n), see Appendix B.6 in the supplementary material;
>
> Q5: See answers in Q3.
>
> Q6: Our data generating process is based on the Dirichlet-Multinomial models, see Appendix B.1 and B.2 the supplementary material;
>
> Q7: We did include the real data example (a commonly used compositional data) in Appendix B.6 the supplementary material;

---

> > ### Author Rebuttal · Reviewer_cpPZ · 2026-04-03
> >
> > No further comments have been provided.

---

### Official Review · Reviewer_Hth6 · 2026-03-11

**Soundness:** 3
**Presentation:** 3
**Significance:** 2
**Originality:** 2
**Overall Recommendation:** 3
**Confidence:** 3

**Summary:**

To address the limitations of existing nonparametric two-sample tests that are primarily designed for Euclidean data and rely on transformations, this study presents a unified framework applicable to both compositional and directional data. The proposed method utilizes a studentized spherical harmonic energy distance-based test integrated with U-statistics theory. By establishing the asymptotic normality of the test statistic, this approach eliminates the need for permutation or bootstrap procedures. Simulations demonstrate theoretical convergence to the limiting distribution, empirical size control, and improved statistical power in specific scenarios, providing an alternative methodology for nonparametric testing in non-Euclidean data analysis.

**Compliance With Llm Reviewing Policy:**

Affirmed.

**Final Justification:**

I’m still not fully convinced that the experiment demonstrates a clear advantage, so I’ll keep my current score.

**Key Questions For Authors:**

see weakness

**Limitations:**

yes

**Strengths And Weaknesses:**

**Strengths:**
1. The proposed method is supported by theoretical guarantees. The asymptotic distribution results are formally established and corroborated by empirical experiments.
2. The paper is well-written, and no obvious notational errors were observed.

**Weaknesses:**
1. The main idea, specifically the approach of transforming compositional data into directional data, is derived from the existing work of Li et al. (2023).
2. The evaluation has limitations, primarily the lack of experiments on real-world datasets. Furthermore, it misses a baseline specifically tailored for directional statistics, such as Maximum Mean Discrepancy (MMD) employing a von Mises-Fisher kernel [1].
   [1] Xu W, Matsuda T. A Stein goodness-of-fit test for directional distributions[C]//International Conference on Artificial Intelligence and Statistics. PMLR, 2020: 320-330.
3. The font size in Figure 3 is too small.

---

> ### Author Rebuttal · Authors · 2026-03-31
>
> In this response, the supplementary material can be found via the following anonymous url "https://anonymous-public-assets.s3.us-west-2.amazonaws.com/30744AppendixB.pdf", the updated codes are available upon request.
>
>
> Response to weakness 1:  At first glance, our paper does seem to share resemblance with Li, Yoon, & Ahn’s JMLR (Li et al.) paper, although that  paper did give us some philosophical inspiration for using RKHS (including spherical harmonics) to study Non-Euclidean Data. However, none of constructions in Li et al. was used in our paper, and their key construction of compositional kernels did not appear in our paper:
>
> (a) Li et al.’s interpreted compositional data as Gamma-invariant directional data through the spread-out procedure,  while our paper argued that such spread-out is not needed;
>
> (b) The key construction of Li et al.’s work is “compositional kernels”, our paper only works with spherical kernels; compositional kernels failed to give the asymptotic normality in Theorem 4.7 (due to its slow growth rate with “p”). Therefore, Li et al.’s methodological constructions do not work for our purpose;
>
> (c) Li et al. gave different connections with the sphere to resolve the boundary issues in compositional data analysis, but boundary issues are not an issue for our purpose, so we only connected compositional data with the first orthant sphere, but such connection was widely used before Li et al.’s work;
>
> (d) Li et al.’s paper did not consider two sample problems;  meanwhile, our paper proposes a new two-sample test for both composition and spherical data and demonstrates simulation studies and real data analysis.
>
> Furthermore, we developed a substantial new framework with RKHS for Non-Euclidean data, and our construction has not appeared in the literature before, stated as follows:
>
> (1) Traditional application of RKHS has infinite dimensionality issues of the function space, which causes difficulty of using high dimensional techniques in RKHS. Similarly we still want to utilize the function space L^2(S^d) (infinite dimensional), which is not an RKHS, and makes the use of high dimensional techniques almost impossible.
>
> (2) The key insight in our work is to decompose the infinite dimensional spaces into Laplacian eigenspace, each of which is a “finite dimensional” RKHS, and thus allowing us to apply high dimensional techniques, which would otherwise be impossible in the infinite-dimensional space. We form the direct sums of these finite dimensional eigenspaces in an asymptotic process to achieve beautiful asymptotic results (the CLT).
>
> In summary, our framework is a combination of the “High-Dimensional problem via eigenspaces” with "asymptotic direct sums,” which is a new paradigm for Non-Euclidean Data Analysis.
>
>
>
> Response to weakness 2:  goodness of fit test is out of the scope for our paper, however, we did experiment with von Mises Fisher kernels, see Table 2 of Appendix B.5 in our supplementary material, which clearly shows that our method has higher power;
>
> Response to weakness 3:  we provide a larger and more legible plot, see Figure 4 in Appendix B.5 in our supplementary material.

---

> > ### Author Rebuttal · Reviewer_Hth6 · 2026-04-02
> >
> > Thank you for your response. To ensure a comprehensive comparison, we would appreciate it if you could provide the Type I error results for MMD using von Mises-Fisher (vMF) kernels. Additionally, could you offer an intuitive explanation as to why the proposed method outperforms the vMF kernels? Furthermore, we would like to see additional results across a broader range of bandwidth parameters for the vMF kernels to better understand their sensitivity.

---

> > > ### Author Response · Authors · 2026-04-06
> > >
> > > Thank you for your further questions, and we address them as follows:
> > >
> > > Q1: Provide the Type I error results for MMD using von Mises-Fisher (vMF) kernels.
> > >
> > > Answer: We have included the Type I error rate for both the vMF-MMD and our method as well in Table 2 and 3 in Appendix B (updated url https://anonymous-public-assets.s3.us-west-2.amazonaws.com/AppendixB.pdf). Please note that more importantly we also included the power analysis as well (and thus also captures the Type II error). Our studentization technique excels at magnifying the distance between the samples when they are close together. Thus, we expect our method to perform well under the alternative hypothesis, but remains well calibrated regarding the Type I error rate. We observed this empirically in our reported results in Tables 2 and 3.
> > >
> > > Q2: An intuitive explanation as to why the proposed method outperforms the vMF kernels:
> > >
> > > Answer: This is an interesting question, as it touches some deeper aspects of our theory.  We list the following reasons on why vMF kernels are not a good fit for our purpose, although they seem to be one of the most reasonable choices:
> > >
> > > (a) With a vMF kernel, an immediate idea is to use this kernel to perform MMD test for spherical distributions, but a key point of our work is to avoid MMD tests, because they suffer from inconsistency issues when two distributions share some traits but differ in other aspects. As a result, we develop studentization techniques for compositional/directional data in Non-Euclidean Settings in this paper;
> > >
> > > (b) As we stated in the introduction of this paper, any MMD method requires that the kernel must be characteristic. Considering that the Hilbert space in our work is $L^2(\mathbb S^d)$, which contains a wide range of both function classes and distribution classes, vMF kernels fail to be characteristic for $L^2(\mathbb S^d)$. The function space $L^2(\mathbb S^d)$ does not admit any reproducing kernels, not to mention characteristic kernels, which motivated us to use spherical harmonic kernels on all Laplacian eigenspaces and construct asymptotic statistics in order to resolve this issue. Therefore, vMF kernels are not ideal to perform MMD-based tests for our function spaces;
> > >
> > > (c) vMF kernels in studentization: if we “orthogonally project” a vMF kernel into each Laplacian eigenspace in $L^2(\mathbb S^d)$, and plug those projected vMF kernels into our studentized statistics in replacement of our spherical harmonics kernels, then an issue would arise: since the projected vMF kernels do not diverge as the projected degree goes to infinity, hence violating Condition (48) (in Appendix A of our paper), so it fails to give an asymptotic normal statistic under the null hypothesis. Therefore, vMF kernels could not be used with studentizations either;
> > >
> > > (d) Last, vMF kernels are non-canonical, as one needs to choose bandwidth parameters, while spherical harmonics kernels have clean expressions through Gegenbauer polynomials, which are purely canonical without any artificial choices of extra parameters.
> > >
> > >
> > > Q3: Additional results across a broader range of bandwidth parameters for the vMF kernels to better understand their sensitivity.
> > >
> > > Answer: In Appendix B (in the updated link: https://anonymous-public-assets.s3.us-west-2.amazonaws.com/AppendixB.pdf), Table 3 and Figure 5 provide an additional experiment to measure the effects of different vMF bandwidth parameters on empirical power with different dimension sizes and sample sizes. We observed is that even when tuning the vMF-MMD bandwidth to be optimal, our test statistic still has higher power.

---

### Official Review · Reviewer_iPxX · 2026-03-13

**Soundness:** 3
**Presentation:** 2
**Significance:** 4
**Originality:** 4
**Overall Recommendation:** 4
**Confidence:** 3

**Summary:**

The paper proposes a studentized spherical harmonic energy distance-based non-parametric two-sample test for compositional data. The authors incorporate U-statistics theory and recent developments of studentization to establish asymptotic normality of the proposed studentized test statistics, avoiding the need for permutation or bootstrap tests. Numerical studies demonstrate the good empirical performance of the proposed test compared to the existing methods. Theoretical advantages of the proposed method of the proposed unifying framework include: no issue of zero caused by the log transformation, and no permutation because of the studentization.

**Compliance With Llm Reviewing Policy:**

Affirmed.

**Final Justification:**

The authors mostly addressed my concerns. Overall, the paper proposed an original solution for an important problem. The presentation is okay, but not clear enough. I think the numerical results can (mostly) support the authors' claim, but it also seems that the experimental part is not very strong and could be improved. And I still maintain my original rating of 4.

**Key Questions For Authors:**

1. There are multiple SPH variants in the numerical studies. And their performance could be quite different. In practice, how to choose which SPH variant to use? Could the author give any guidance on this?

2. The paper claims that the performance advantage of the proposed method shines when the means agree and the distributional differences occur in higher-order moments. Although the method can detect differences in higher-order moments, I think that the order of the moments that can be detected should still be restricted by the sample size. Is this true? Could the authors design and provide some numerical studies that show the ability or the limit of the difference order detected by the proposed method?

**Limitations:**

1. The authors should have some discussion on how to choose between different SPH tests.

2. And it is also interesting to see SPH or similar methods extended to high-dimensional low sample size (HDLSS) cases, which the authors also mentioned in the Discussion section.

**Strengths And Weaknesses:**

Strengths：
1. The paper proposes a novel unifying framework for two-sample testing for compositional data utilizing harmonic energy distance, and solves critical issues in existing methods, including log-transformation methods and energy distance methods on Euclidean space.
2. By studentization, the authors derive the asymptotic null distribution rigorously using U-statistics theory, thus avoiding the need for permutation tests.
3. The empirical studies show good numerical performance of the proposed method.

Weaknesses：
1. There can be multiple variants of the proposed method, but there is not much clear guidance on how to choose between different variants.
2. The content/derivation in the paper is quite adequate, but the overall presentation/organization of the paper can be improved to be clearer.

---

> ### Author Rebuttal · Authors · 2026-03-31
>
> In this response, the supplementary material can be found via the following anonymous url "https://anonymous-public-assets.s3.us-west-2.amazonaws.com/30744AppendixB.pdf", the updated codes are available upon request.
>
> Response to Key Question 1: While we observe that our test statistic performs better with the L2 normalization to map from the simplex to the first orthant of the sphere when the dimension is small and the signal lies in the dominant taxa (μ abundant alternative), the most robust choice across varying dimension, sample size, and type of alternative is to select the version where we use the square root transformation. The heuristic of choosing p ∼ log(d) works well. Thus as a practical prescription for practioners we recommend to use SPH-√-pn where p ∼ log(d). In our experiments SPH-√-p4 was the most robust choice.
>
> Response to Key Question 2: Since the null hypothesis in our setup is such that the distribution of each sample is the same, our method detects differences at the “distributional level” and not solely differences in the mean. Thus, our method is applicable even if the means are equal, and it is sensitive to the shapes of the distributions, which is determined by the higher moments and other features. For instance, our test statistic is able to distinguish distributions on the sphere even when the means are identical, but higher order moments differ, see Figure 4 in Section B.5 in our supplementary material.
>
> Regarding sample sizes, our studentized statistics “magnify” the kernel energy distance (KED) by dividing its variance; indeed in Proposition 5.2 (page 7), the term of c_{m,n} in the variance goes to zero as sample size grows, which does magnify the kernel energy distances (KED). This magnification is significant even with small sample sizes, although larger sample sizes magnify the KED even more. In short, a decent sample size is needed, but a small sample size is not as restrictive in practice. We included a real data analysis with small sample sizes in Section B.6.2, and also Table 2 in Section B.5  in our supplementary material.
>
>
> Response to Limitation 1: see our answer to Key Question 1.
>
> Response to Limitation 2: an excellent point, HDLSS is an important new topic which we will address in a separate and forthcoming paper, as it requires new theoretical work outside of the scope of this article. Our plan is to embed spherical domains into RKHS’s of a fixed degree while varying the underlying dimensions of spheres, and develop two sample tests on the “embedded measures” (a.k.a. Varifold measures) over finite dimensional Banach spaces. Our real data analysis in Section B.6.2 (in our supplementary material) is an instance of HDLSS scenario, where the sample size is much smaller than the dimension, and our method still performs well.

---

> > ### Author Rebuttal · Reviewer_iPxX · 2026-04-01
> >
> > Thank you for your response and clarification. The response partially resolved my concerns. Specifically, for Question 2, I'm more concerned with the order of the moments (e.g., 2nd or 4th order moments?) that the proposed method can detect, since detecting higher-order moment differences is a main claim and advantage of the proposed method. While the empirical results did show that the moment difference higher than order 1 (mean) can be detected, they gave relatively little hint to my concern. I asked the question because I felt that the sample size needed by the proposed method would grow exponentially with the order of the moment difference. For example, if for two distributions with 1st order difference, you need n samples, then for two distributions with 4th order difference (but same 1st and 2nd orders) of a comparable magnitude, you would need n^4 samples for the proposed test to achieve a comparable power. I hope that the authors can do some further theoretical and/or empirical analysis on this issue, since this can be quite important for the applicability of the proposed method.

---

> > > ### Author Response · Authors · 2026-04-06
> > >
> > > We sincerely appreciate your questions and address your concerns below:
> > >
> > > Q1: The order of the moments (e.g. 2nd or 4th moments) in two sample tests:
> > >
> > > Answer: This is a subtle point, which we address through the following perspectives:
> > >
> > > a. In two sample tests, there are univariate cases, and multivariate cases (dimension d >1); in multivariate cases, situations also differ for distributions on Euclidean and Non-Euclidean spaces. In univariate cases, higher-order moments (if they exist) are scalars, for which it is reasonable to address the higher-order moment differences.
> > >
> > > b. In multivariate settings, higher order moments are no longer scalars, but “higher-order tensors.” The second moments of multivariate Gaussians are covariance matrices, whose two sample problems were studied via random matrix theory (see [2]).  Higher moments for multivariate Gaussians involve higher-order tensors, which is more intricate than covariance matrices. Hence, we usually do not find higher moment differences beyond the first two moments for multivariate cases in the literature.
> > >
> > > c. In compositional/directional statistics, not only does it fall into multivariate studies, but also in “Non-Euclidean” studies, for which it is nontrivial to construct parametric distributions, although progress has been made by imitating multivariate Gaussians (e.g., Spherical Kent Models) or exponential families (e.g., Compositional Exponential Models). In those models, we can discuss the “mean” and “covariance-tensor” in a sense, but higher-order moment tensors are generally intractable, so Non-Euclidean higher-order moment tensors have also not been addressed in the literature either.
> > >
> > > d. In order to distinguish distributions (on higher order moment level) in multivariate \& Non-Euclidean cases, a practical strategy is to develop nonparametric methods, as we studied in our paper.
> > >
> > > e. Non-parametric two sample tests were based on MMD, but MMD methods are inconsistent, especially when two distributions have the same features in some ways but differ in other aspects (e.g. higher-order moments). The reason of those inconsistencies is due to the fact that two distributions are too close to each other, therefore, we introduce studentizations to “magnify” energy statistics, even for distributions that differ only on higher-order moment tensors.
> > >
> > > In conclusion, our paper establishes the first complete nonparametric two sample framework for compositional and directional statistics that is sensitive to distributional differences beyond the first moment (e.g. mean). While other works have focused only on mean differences (see [1] and [3]) or nonparametric methods just for the circular case (see [4]), our framework is capable of detecting higher-order distributional differences for arbitrary dimensions at a practical sample size (please see the next answer below).
> > >
> > > Q2: Do we need a larger sample size to tell the differences of distributions caused by higher-order moments?
> > >
> > > Answer: Our studentization magnifies the energy statistics by dividing it by its variance component, but it sometimes gives the misleading impression that the only “magnifying” factor comes from $c_{m,n}$, so to distinguish distributions with subtler differences one needs smaller $c_{m,n}$ (which therefore calls for larger sample). However, this is not the case, because another factor in the variance component is the (kernel) squared distance covariance (sdCov), (sdCov can be roughly viewed as the squared distance of two distributions), so sdCov also shrinks if two distributions are closer; in particular, if two distributions only differ on higher-order moment tensors, the sdCov factor decreases automatically, without relying on the increase of sample sizes to magnify the energy statistics.
> > >
> > > Table 2 in Appendix B (updated url:https://anonymous-public-assets.s3.us-west-2.amazonaws.com/AppendixB.pdf) presents how different sample sizes and dimensions affect the empirical power for the two distributions’ $\kappa_1/\kappa_2=1.75$ with the same mean, which implies the second and higher moments are different. As can be seen, our test outperforms the vMF test even at a small sample size of 25, which becomes more widened as the dimension increases.
> > >
> > >
> > >
> > >
> > >             [1] Cao, Y., Lin, W., & Li, H. (2018). Two-sample tests of high-dimensional means for compositional data. Biometrika, 105(1), 115-132.
> > >             [2] Li, Haoran, Alexander Aue, Debashis Paul, Jie Peng, and Pei Wang. "AN ADAPTABLE GENERALIZATION OF HOTELLING’S $T^²$ TEST IN HIGH DIMENSION." The Annals of Statistics 48, no. 3 (2020): 1815-1847.
> > >             [3]Li, D., Xue, L., Yang, H., & Yu, X. (2025). Power-enhanced two-sample mean tests for high-dimensional microbiome compositional data. Biometrics, 81(2), ujaf034.
> > >             [4]Jammalamadaka, S. R., Guerrier, S., & Mangalam, V. (2021). A two-sample nonparametric test for circular data–its exact distribution and performance. Sankhya B, 83(Suppl 1), 140-166.

---

### Decision · Program_Chairs · 2026-04-30

**Decision:**

Accept (regular)

**Comment:**

This paper builds a two-sample test for compositional data. They propose a studentized spherical harmonic energy distance-based two-sample test and establish asymptotic normality of the test statistics under the null hypothesis.
All the reviewers agreed that the subject studied is timely and important. They also agree that the method proposed is correct, elegant, and novel.
The main criticism concerned (1) the presentation, which some reviewers felt could be improved, but they also agreed that it was already "ok" and (2) the experimental evidence, notably a lack of comparison with MMD methods. During the rebuttal period, this was clarified, and the authors highlighted those comparisons. Two out of three reviewers are convinced. One reviewer still thinks that empirical evidence could be stronger.
However, overall this seems to be a solid paper with a good theoretical basis. I therefore recommend acceptance.